



# New age constraints for glacial terminations IV, III, and III.a based on Western Mediterranean speleothem records

Judit Torner[1], Isabel Cacho[1], Heather Stoll[2], Ana Moreno[3], Joan O. Grimalt[4], Francisco J. Sierro[5], Hai Cheng[6], and R. Lawrence Edwards[7]

[1]GRC Geociències Marines, Dept. de Dinàmica de la Terra i de l'Oceà, Universitat de Barcelona, Barcelona, 08028, Spain
[2]Department of Earth Sciences, ETH Zúrich, 8092, Switzerland
[3]Departamento de Procesos Geoambientales y Cambio Global, 639/, CSIC, 50059 Zaragoza, Spain
[4]Institute of Environmental Assessment and Water Research (IDAEA-CSIC), Barcelona, 08034, Spain
[5]Departmento de Geología, Universidad de Salamanca, Salamanca, 37008, Spain
6Institute of Global Environmental Change, Xian Jiaotong University, Xian 710049, China
[7]Department of Earth Sciences, University of Minnesota, MN 5545, USA

*Correspondence to*: Judit Torner (j.torner@ub.edu)

**Abstract.** The full understanding of climate feedbacks responsible for the amplification of deglaciations requires robust chronologies for these climate transitions, but, in the case of marine records, radiocarbon chronologies are possible only for the last glacial termination. Although the assumed relationships between the marine isotopic record and the orbital parameters provide a first order chronology for the previous terminations, an independent chronological control allows the relationships between orbital forcing and the climate response to be assessed over multiple previous terminations. Here we present new

geochemical records of Marine Isotope Stages 11 to 7 from a western Mediterranean speleothem, establishing a new long terrestrial climate record for this region. Its absolute U/Th dates provide an exceptional chronology for the glacial terminations IV, III, and III.a. The onset of these three glacial terminations was marked by rapid $\delta^{18}$O depletions, reflecting ocean freshening by ice melting, thus providing an excellent tie point for regional marine records also sensitive to such freshening. These new chronologies reveal an earlier onset of the deglacial melting for the TIV and TIII.a in contrast to the generally accepted marine

chronologies and indicate that the duration of these deglaciations was variable, with TIV particularly longer (~20 kyr). This study also supports that the onset of deglacial melting always occurred during declining precession index while a nonunique relation occurred with the obliquity parameter.

## 1 Introduction

Glacial-interglacial transitions exemplify the highest amplitude of climate changes along the Late Quaternary and involve

intense and fast processes of ice-sheet melting that led to rapid sea level rise (PAGES, 2016). Although increasing north hemisphere summer insolation (NHSI) is considered the main driver for deglaciations, internal feedbacks from the Earth's climate system were crucial to trigger glacial terminations. (Cheng *et al*., 2009; Drysdale *et al*., 2009; Tzedakis *et al*., 2012;



Barker *et al.,* 2021). The consequences of the insolation rise depended on the actual extent of the ice-sheet build-up during the preceding glacial period, determining the main features of terminations (Tzedakis *et al.,* 2017). Ice-sheet instability and intense

events of massive icebergs released into the N Atlantic, named Henrich-like events, are considered characteristic features of deglaciation (Barker *et al*., 2015, 2019, 2021; Hodell *et al*., 2013, 2023) and have been well-documented through ice rafting debris (IRD) records in the N Atlantic (Hodell *et al*., 2008; Barker *et al*., 2011). The consequent North Atlantic freshening caused by ice melting became the trigger for powerful feedback processes inducing glacial terminations. The freshwater discharge allowed surface water stratification and provoked the displacement of the deep-water formation, disturbing the

strength of the Atlantic Meridional Overturning Circulation (AMOC) (Rahmstorf 2002). This, in turn, changed the interhemispheric heat transport, inducing a rapid cooling in the North Atlantic region, whereas Antarctica gradually warmed. The asymmetry in the hemispheric heat transport (the bipolar-seesaw) and its influence on the atmospheric $CO_2$ rise was also a fundamental requirement to enter into a deglaciation (Cheng *et al*., 2009; Barker *et al*., 2011, 2019, 2021; Tzedakis *et al*., 2012). Thus, variations in the AMOC and greenhouse gases explain much of the variability in global climate during

deglaciations (Clark *et al*., 2012). In parallel to the weakening of the AMOC, albedo changes due to the reduction of continental ice sheet extent operated as secondary internal feedback or amplifier. At the end of deglaciations arises the characteristic abrupt warming leading to the onset of interglacial periods but, the sea level highstand marking the interglacial conditions, occurred systematically a few thousand years after the maximum in NH summer insolation due to the slow response of the ice sheets to insolation (PAGES, 2016). Several warm marine isotopic substages are considered interglacial acmes or peaks of optimum

climate conditions (the MIS 7a-c, 7e, 9e, and 11c within them). However, temporal and geographical asynchronies in the precise definition of their boundaries between marine and terrestrial records or even different proxies within the same archives pose a challenge to resolving the feedbacks acting in these transitional periods (Shackleton, 1969 and 2003; Sanchez-Goñi *et al*., 1999).

Astronomically-tuned marine sediment records provided fundamental information to understand the evolution of the continental ice sheets in the past. Globally averaged $\delta^{18}O$ stacks from benthic foraminifera, reflecting changes in ice-sheet build-up and deep ocean temperature, have provided a global chronological framework for correlating marine sediment sequences, and the marker is its global signal driven by changes in the ice-sheet build-up. (Imbrie *et al*, 1984; Martinson *et al*., 1987; Lisiecki and Raymo, 2005). However, the chronology applied to such stacks is based on the assumption of a stationary

relationship between insolation and ice volume changes and does not account for suborbital climate influence in the time sequences of each glacial termination (Hodell *et al*., 2015). While also reflecting ice volume cycles, the seawater oxygen isotopic signal ($\delta^{18}O_{sw}$) from planktonic records can be overprinted by regional temperature or hydrographic changes challenging its use as an isotopic chronostratigraphic tool (Rohling *et al*., 2015). Hence orbitally dependent marine isotopic chronologies are limited in resolving precise age constriction of key climate periods, such as glacial-interglacial transitions,

complicating any interpretation in terms of orbital forcing.



A growing number of studies use absolute dating from speleothems to anchor the marine records and achieve astronomically independent age models (e.g. Grant *et al*., 2012; Cheng *et al*., 2016). Speleothems with an unambiguous climatic interpretation of their proxies spanning continuously several glacial-interglacial cycles are restricted to a few studies such as those recording the Asian monsoon in China (Yuan *et al*., 2004; Cheng *et al*., 2006, 2009 and 2016), or those focused on America (Cruz *et al*.,

2005 and 2009; Moesley *et al.,* 2016). Superimposed on a strong precessional $\delta^{18}O$ signal forced by regional insolation and sea surface temperature gradients the precisely dated speleothem records from Sanbao cave, in China, show positive $\delta^{18}O$ anomalies interpreted to reflect changes in the position and timing of monsoon rainfall coincident with the weakening of the AMOC in the North Atlantic (Wang, *et al.,* 2001; Cheng *et al*., 2009; Cai *et al.,* 2015; Barker *et al*., 2021; Jian *et al*., 2022). If AMOC weakening is linked to glacial meltwater release in the North Atlantic, these records may therefore provide indirect

indicators of the chronology of glacial terminations. However, the distinctive freshening signature of the glacial terminations and therefore the specific time sequence of its events cannot be recorded in these speleothems located far from the N Atlantic water sources. Hence, well-constrained speleothem records capable of recording the N Atlantic deglacial features through its isotopic signature would be advantageous to establish independent chronology on North Atlantic marine sequences and clarify the forcings of deglaciations. In this respect, the first high-resolution speleothem records successfully used to tune marine

records were collected in the NW Iberia peninsula and documented rapid N. Atlantic ice sheet melting spanning the two ultimate glacial terminations (TI and II) at millennial-centennial timescale (Stoll *et al.,* 2022).

Regarding the Mediterranean region, available speleothem records that can be potentially used for marine alignments are mostly restricted for the two most recent deglaciations (e.g. Bar-Matthews *et al*., 1999, 2000 and 2003; Drysdale *et al*., 2009; Tzedakis *et al*., 2018). In the western Mediterranean basin, only the Ejulve speleothem record from the eastern Iberian Peninsula detected millennial-scale events related to AMOC slowdowns at TIII. While speleothems spanning TIV and TV remain elusive in the Mediterranean context (Kaushal *et al*., in review 2024). This study presents a new speleothem record from Minorca Island in the western Mediterranean (Balearic Islands) (Fig.1), which grew continuously through the period between marine isotope stage 11 to 7 (MIS 11-7), thus covering two glacial-interglacial cycles. A geochemical multiproxy

approach has been applied to further investigate climate variability with a special focus on the glacial terminations TIV, TIII, and even the additional TIII.a. First insights into the existence of this extra glacial termination were previously reported in the Chinese speleothem records, which argue that the TIII.a shows patterns of events equivalent to other glacial terminations despite being amid MIS 7 (Cheng *et al*., 2009). The studied deglaciations present different intensities and durations and open an opportunity to evaluate the chronology of orbitally tuned paleoclimatic records.



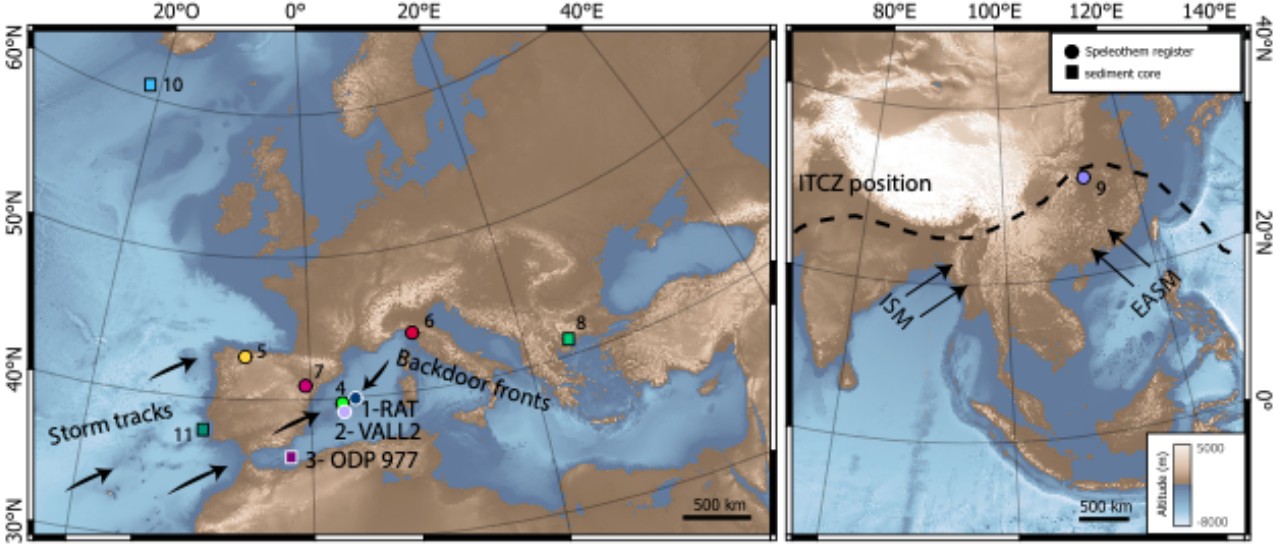

**Figure 1: Map showing the location of this study records: 1) Murada cave, in Menorca Island, where the RAT speleothem was recovered. 2) Vallgornera cave in Mallorca Island, where the VALL 2 speleothem was recovered. 3) The marine sediments from the ODP 977 site. Other sites discussed in the text are also situated: 4) Campanet cave on Mallorca Island (Dumitru *et al.*, 2018). 5) NISA and Garth speleothem records from north Iberia caves (Stoll *et al.*, 2015 and 2022). 6) Corchia cave (Drysdale *et al.*, 2004 and Tzedakis *et al.*, 2018). 7) Ejulve cave (Pérez-Mejías *et al.*, 2017). 8) The Tenaghi Philippon arboreal pollen record (Tzedakis *et al.*, 2006). 9) Sanbao speleothem record (Cheng et al., 2009; 2016). 10) The ODP 983 site (Barker *et al.*, 2015, 2019, 2021). 11) The MD01-2444 marine core (Tzedakis *et al.*, 2018; Hodell *et al.*, 2013). The map shows the dominant atmospheric circulation that brings moisture to Menorca (via Mediterranean backdoor cold fronts and winter storm tracks). While moisture in Asia is related to the dynamics of the East Asian Summer Monsoon (EASM) and the Indian Summer Monsoon (ISM), which depends on the Intertropical Convergence Zone (ITCZ). The map shows the modern average position of the ITCZ in summer.**

## 2 Material and Methods

### 2.1 Hydrological cave settings

The speleothem called Ratpenat (RAT) was collected from inside *Murada* cave, in Minorca, one of the Balearic Islands in the western Mediterranean Sea. The Mediterranean region has important climate contrasts among different nearby areas because is a transitional zone between subtropical and midlatitude temperate regimens (Lionello, 2012). While the northern part of the Mediterranean region is strongly linked to midlatitude variability, mostly characterized by the North Atlantic Oscillation, the southern and eastern parts are exposed indirectly to the influences of the Inter-Tropical Convergence Zone dynamics (ITCZ). The two main origins of the precipitation on the western Mediterranean Sea are the Mediterranean Sea itself and the tropical–subtropical North Atlantic corridor via the dominant eastward atmospheric circulation, leading storm tracks reaching the Mediterranean (Gimeno *et al.*, 2010; Nieto *et al.*, 2010; Krklec and Domínguez-Villar 2014; Dumitru *et al.*, 2017; Moreno *et al.*, 2021). However, the major source of moisture for precipitation in the Mediterranean surrounding land area has been identified as the basin itself (Schicker *et al.*, 2010). Particularly in the Balearic Islands location, the moisture is provided by





evaporation over the Mediterranean because of the occurrence of backdoor cold fronts related to easterly advection leading to heavy precipitation in autumn (Millan *et al*., 2005; Moreno *et al.,* 2021). According to the Köppen climate classification, the Balearic Islands present temperate climates (Csa) with seasonal rainfall, intense storms in autumn, and dry/hot summers, which cause the lack of permanent watercourses. Minorca is characterized by an irregular regime of ravines, especially in autumn when the intense storms cause sporadic high flows. The *Murada* cave is located close to the coast and at little altitude, 80 m a.s.l., in the southwest area of Minorca (39°57'58''N; 3°57'53''E) (Fig. 1). This cave is the largest in the *Barranc d'Algendar*, which is a canyon formed by a ravine cutting the Miocene calcarenite rocks where a karst was developed. Nowadays this cave represents a fossil level of the karst drainage conduit. The RAT speleothem was found broken in two pieces which were matched in the cave.

### 2.2 Geochemical analyse

#### 2.2.1 Geochronology of RAT

In order to determine the speleothem absolute ages, RAT was longitudinally cut with a saw taking into account the morphology and the growth axis. RAT is a 50 cm long speleothem broken at the top; a 2cm calcite piece was missing when the sample was found in the cave. It presents a prominent orange-brown layer at 14 cm from the tip and white layers at 19 and 33 cm. The sampling for measuring uranium and thorium isotopes was designed considering the layering to detect possible hiatuses and changes in growth rates (Fig.2). A total of 41 samples were milled with a 2 mm diameter tungsten carbide micro-drill under a laminar airflow cabinet to prevent contamination. About 100-200 mg of carbonate powder was obtained for each sample. The chemical procedure for isolating uranium and thorium elements was performed in the ultra-clean laboratory at the University of Minnesota and the Xi'an Jiaotong University, applying the methodology previously described by (Shen *et al*., 2002; Cheng *et al*., 2009) (Appendix A). Several blanks were performed routinely for each laboratory sampling set. The radiogenic isotopes of U and Th were analyzed using a parallel ion-counting Multi Collector Inductively Coupled Plasma Mass Spectrometry (MC-ICP-MS) with a *Finnigan Neptune* model at both universities. The results of these analyses and the absolute dates are detailed in Table A1. The depth-age model was performed with 39 radiometric dates using the R software and the StalAge package (Scholz, 2011), providing a consistent chronology with only a single reversal exceeding the dating uncertainty; absolute uncertainties are smaller in the youngest part of the speleothem. The speleothem growth rates fluctuate between 0.07-1.7 cm/kyr (Fig.2).



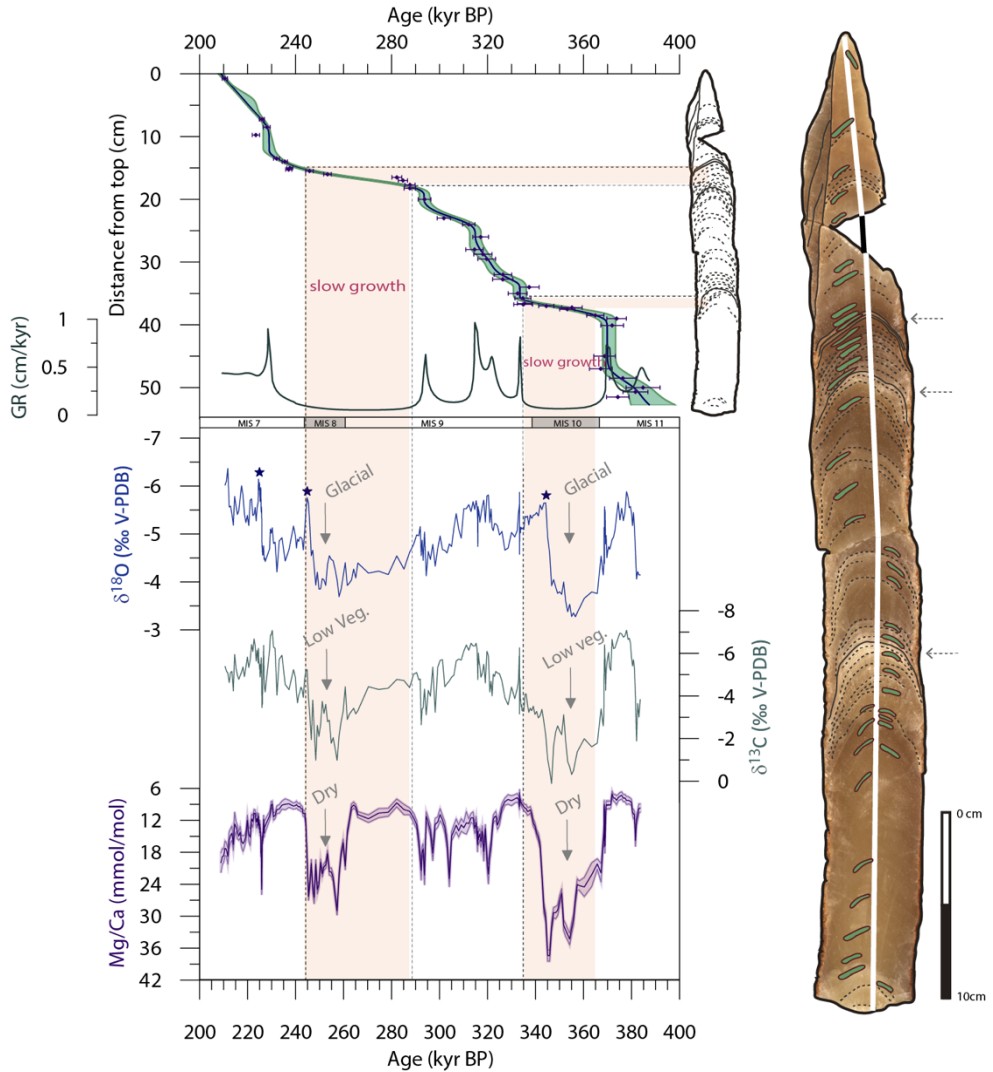

Figure 2: On the right, the image of the RAT speleothem with the position of the sampled radiometric dates used for the age model and the growth axis path used for the geochemical analyses (white line). At the top, the age model acquired by StalAge (dark blue line) and the original U/Th dates for the RAT speleothem (dark blue dots) plotted with the associated errors (error bars and green area). RAT growth rates are shown in dark green. In the lower panel the δ¹⁸O, δ¹³C, and Mg/Ca results. Note that the three parameters are plotted with a reversal axis. The stars highlight negative δ¹⁸O anomalies mentioned in the main text. The pink bars highlight slow growth rate periods, and the dashed grey line points to major color changes of the speleothem layers also noted as dashed grey arrows in the image.

## 2.2.2 Geochemical measurements in RAT

Stable isotopes and trace elements have been concurrently measured in the RAT speleothem. δ¹⁸O, δ¹³C, and Mg/Ca ratios have been estimated and compared in detail to evaluate their paleoclimatic signals. A total of 315 samples were drilled using a 0.5 mm diameter tungsten carbide dental burr at 2.5 mm resolution along the speleothem growth. Samples are more closely spaced (every 0.5 mm) during glacial terminations to increase the temporal resolution because of low growth rates in these



intervals. The $\delta^{18}O$ and $\delta^{13}C$ measurements were performed with 50 µg of carbonate powder and were accomplished on a Finnigan-MAT 252 mass spectrometer (IRMS) at the Scientific and Technological Centres of the University of Barcelona (CCiT-UB). This mass spectrometer is coupled in a single acid bath CarboKiel‑III carbonate preparation device to convert solid carbonate samples into simple gas ($CO_2$) before entering into the ion source of the IRMS. Analytical uncertainties were obtained utilizing two in-house carbonate standards that were calibrated to NBS-19 international standard (Coplen, 1996). The uncertainties were ≤ 0.04 ‰ VPDB for $\delta^{13}C$ and 0.08 ‰ VPDB for $\delta^{18}O$.

Around 3 mg of carbonate powder was used to measure Ca, Mg, and other trace elements such as Ba and Sr to obtain element/calcium ratios. The powder was directly dissolved with 3 ml of ultra-pure 2% $HNO_3$ (with Rh and Sc as internal standards) and centrifuged to prevent possible lithic particles or impurities in the solution before the analyses. These samples were analyzed employing inductively coupled plasma mass spectrometry (ICP-MS), a Perkin-Elmer Elan-6000 model, at the CCiT-UB. Possible contamination was controlled by means of chemistry blank analyses on random days. An in-house high-purity standard solution was measured routinely every four samples to control the accuracy of the measurements. According to the in-house standard analysis, the external reproducibility obtained for the speleothem analyses was 0.86% (%, 2-sigma). The records present an overall variation of 3.2‰, 6.3‰ and 24.6 mmol/mol for $\delta^{18}O$, $\delta^{13}C$, and Mg/Ca respectively.

### 2.2.3 Planktic foraminifera $\delta^{18}O$ in ODP 977

ODP Site 977 was retrieved during LEG 161 of the Ocean Drilling Program on board the *JOIDES Resolution* cruise. It was located halfway between the Spanish and Algerian coasts, in the eastern sub-basin of the Alboran Sea, at 36°01.92'N and 1°57.32'W, and from a water depth of 1984 m b.s.l. (Fig.1). This study extends the oxygen isotope data obtained in previous studies of this sediment core (Martrat *et al.,* 2004 and 2007; Gonzalez-Mora *et al.,* 2008; Torner *et al.*, 2019; Azibeiro *et al.*, 2021). The ODP 977 age models of all these different studies are mostly based on the original age model from Martrat *et al.*, 2004 which is established on the correlation between its *Globigerina bulloides* $\delta^{18}O$ curve and the SPECMAP stacked curve (Martinson *et al.*, 1987). However, in Torner *et al.*, (2019) some additional tie points were used to align the TII with other Mediterranean cores radiometrically tuned with speleothems. Whereas Azibeiro *et al.*, (2021) adjusted the age model along the TV by aligning its alkenone-based SST record to North Atlantic mid-latitude temperature records.

Stable isotopes on planktic foraminifers were performed on handpicked specimens of *G. bulloides*. Shells were crushed between two glass plates to open the chambers and favor the following clay removal. Afterward, samples were cleaned with methanol and sonicated for a few seconds. The supernatant solution was removed with a micropipette and the samples were finally dried. The measurements were accomplished by isotope ratio mass spectrometry at the CCiT-UB using the same analytical method that was previously explained for speleothems analyses.


## 3 Results and Discussion

### 3.1 The climate signal in speleothem RAT

Mg/Ca ratios and stable isotope in the speleothem RAT reveal climate variability in the western Mediterranean region from 387 to 209 ka BP, covering continuously an exceptional long-time-interval (178 kyr) that includes three glacial Terminations (TIV, TIII and TIII.a). According to the obtained depth-age model, the glacial conditions of the MIS 10 and MIS 8 are characterized by slower growth rates and high $\delta^{18}O$, $\delta^{13}C$, and Mg/Ca (Fig. 2). While the warm marine isotopic stages MIS 11, 9, and 7 feature light isotopes and low Mg/Ca ratios.


The Mg/Ca ratio in speleothems is sensitive to variations in the precipitation/evaporation balance reflecting changes in the amount of water infiltrating into the karst system. Frequently, arid conditions may increase the water residence time and the prior calcite precipitation (PCP) along the groundwater path, as well as reduce both drip and growth rates (Fairchild *et al.,* 2000). During periods of low flow, PCP can preferentially remove Ca from the drip waters, consequently elevating the trace

element/Ca ratio of all incompatible elements in the dripwaters, such as Mg/Ca, Sr/Ca, and Ba/Ca. When the partitioning coefficients of Mg, Sr, and Ba in stalagmite calcite are relatively stable, the speleothem Mg/Ca, Sr/Ca, and Ba/Ca reflect this dripwater PCP signature. The RAT Sr/Ca and Ba/Ca records show similar temporal variations as Mg/Ca variability (Appendix Fig. B1), suggesting that all three elements are dominated by a common process of PCP. The magnitude of change in Mg/Ca is much larger than would arise due to temperature-driven changes in Mg partitioning, which in any case would elevate Mg

incorporation during warm interglacials, in contrast to the observed pattern. PCP can also be enhanced by higher dripwater saturation states even with constant drip rates (Stoll *et al.*, 2012), but the observation of maximum PCP during cold glacial periods characterized by lower respiration rates, lower soil $CO_2$ and likely lower initial dripwater saturation suggests that elevated glacial PCP is caused by slower drip rates and driven by lower infiltration and lower precipitation-evapotranspiration. Speleothem $\delta^{13}C$ can also reflect hydroclimate variations. The initial dripwater $\delta^{13}C$ is higher when soil $pCO_2$ is low due to

lower soil autotropic and heterotrophic respiration rates, and many mid-latitude speleothem studies interpret positive $\delta^{13}C$ shifts as reflecting reduced soil/vegetation activity (Genty *et al.,* 2013; Moreno *et al.,* 2010; Bartolomé *et al.,* 2015; Perez-Mejias *et al*., 2017). Both dry conditions and cold temperatures can depress soil respiration rates and contribute to higher initial dripwater $\delta^{13}C$. Additionally, slowed drip rates during dry periods may contribute to greater PCP, which causes dripwater to evolve to higher $\delta^{13}C$. Thus, via two processes, dry conditions may contribute to higher speleothem $\delta^{13}C$. In the RAT record,

while cold temperatures likely depressed soil respiration during glacials and contributed to the high $\delta^{13}C$, the general co-variation between Mg/Ca and $\delta^{13}C$ suggests that the high glacial $\delta^{13}C$ also reflects dryer conditions contributing to low soil $CO_2$ and slower drip rates favouring more extensive PCP. In a speleothem record from Mallorca, also a Balearic Island, high $\delta^{13}C$ was similarly interpreted to reflect low biogenic $CO_2$ productivity in soil and/or a short water residence time, reflecting dry conditions (Dumitru *et al.,* 2018). The RAT-$\delta^{13}C$ signal shows a smoothed signal with respect to the Mg/Ca during glacial

terminations, suggesting a slower response of the vegetation compared with precipitation rises (Fig.2). The MIS 11, 9, and 7





were characterized by enhanced precipitation conditions that led to higher soil respiration activity under more extensive vegetation.

The $\delta^{18}$O RAT record is marked by higher $\delta^{18}$O during glacials and lower $\delta^{18}$O during interglacials (Fig.2). On a global scale,
the retreat of large continental Northern Hemisphere ice volume and release of meltwater lowers the $\delta^{18}O_{sw}$ in the surface ocean. This lower $\delta^{18}O_{sw}$ causes a corresponding decrease in the $\delta^{18}$O of rainfall and the drip waters from which the speleothem is formed, lowering the speleothem $\delta^{18}$O. Over the most recent glacial cycles, the North Atlantic experiences a greater amplitude change in $\delta^{18}O_{sw}$ than the globally averaged surface ocean (Waelbroeck *et al*., 2014). Whereas in some regions, major changes in atmospheric circulation patterns cause additional larger changes in rainfall $\delta^{18}$O which significantly exceed
the effect of the surface $\delta^{18}O_{sw}$. In coastal regions of the North Atlantic, the change in surface ocean isotopic composition is a dominant component of speleothem $\delta^{18}$O variations over glacial terminations. Comparison of independently dated marine $\delta^{18}O_{sw}$ with speleothem $\delta^{18}$O records from NW Iberia showed that the speleothem $\delta^{18}$O matched the timing and amplitude of the $\delta^{18}O_{sw}$ shift in the eastern North Atlantic during Termination I (Stoll *et al.,* 2022). Furthermore, several studies demonstrate that the N Atlantic $\delta^{18}O_{sw}$ signal, changed by those freshwater inputs, was transferred into the Mediterranean Sea (Cacho *et*
*al*., 1999; Sierro *et al*., 2005; Jiménez-Amat and Zahn 2015; Marino *et al*., 2015). Over Termination I, the North Atlantic $\delta^{18}O_{sw}$ decreased rapidly around 16 ka and between 12 and 10 ka, and these two shifts are also recorded in the Alboran Sea as well as around Menorca (Appendix Fig. C1). Consequently, we propose that the RAT $\delta^{18}$O records the change in $\delta^{18}O_{sw}$ in the moisture source area for the precipitation, encompassing both, global changes of the $\delta^{18}O_{sw}$ at a large scale, and superimposed, the freshening signal transmitted to the Mediterranean Sea during deglaciations. This is also supported by the evaluation of the
isotopic composition of rainfall waters and cave dripwaters in the closer Balearic Island, Mallorca, which points to the western Mediterranean as the dominant source for moisture uptake in coastal caves (Dumitru *et al*., 2017). The cave location of this study, close to the coast and at a low altitude above sea level, should minimize distillation processes, and therefore RAT $\delta^{18}$O captures the signal of the moisture source, surface water mostly from the Western Mediterranean.

### 3.2 Mediterranean precipitation modulated by precession

The RAT Mg/Ca record shows a clear precession pattern. Low Mg/Ca, indicating enhanced precipitation, coincides with precession minima which are times of high north hemisphere summer insolation (NHSI) (Fig.3, pink bars), and vice versa for drier periods. The most humid periods are recorded during the strong precession minima of marine isotopic sub-stages MIS 9e and 9a, and MIS 7e. MIS 11a shows similar humidity despite a moderate precessional minimum. The low resolution of the Mg/Ca record during the MIS 9a precludes us from evaluating its duration and structure. In the better resolved interglacial
acmes MIS 9e and MIS 7e, we estimate a duration of ~10 kyr. These orbitally paced changes in Mediterranean precipitation are explained because minima precession led to higher seasonality (high NHSI-low NHWI), which intensified wintertime storm tracks and increased the Mediterranean precipitation. Additionally, higher air-sea temperature gradients enhanced





evaporation from the Mediterranean basin itself, leading to greater local convective precipitation (Kutzbach et al., 2014; Wagner *et al*., 2019; García-Alix *et al.,* 2021).


This new uninterrupted record covers enough time length to confirm the precession imprint for the first time in a Mediterranean speleothem. Other European speleothems have also recorded enhanced precipitation during precession minima of the studied interglacials, however, they provide shorter records. The iconic $\delta^{18}$O record of the Antro del Corchia cave, from Italy, shows periods of greater humidity in line with those recorded in RAT during the MIS 9. However, higher U/Th date uncertainties

could account for some chronological discrepancies (Fig.3). The increment in moisture availability in western Italy was attributed to regional warm SST that led to enhanced evaporation (Drysdale *et al.,* 2004). Carbon isotope records from other European speleothems are more complex to interpret without accompanying trace elements, but the trends are consistent with warmth and or/ greater humidity during MIS 7e. The $\delta^{13}$C speleothem record from the Ejulve cave, in the north-eastern Iberian Peninsula, shows lower values during the MIS 7e (Fig. 3) and is interpreted as a proxy of vegetation productivity (Pérez-Mejias

*et al.,* 2017). Other speleothem records from the Austrian Alps and Sardinia also show more negative $\delta^{13}$C during MIS 7e which may reflect warming or more humid conditions responding also to changes in the North Atlantic realm (Spötl *et al.,* 2008; Columbu *et al.,* 2019; Wendt *et al.,* 2020).





**Figure 3:** The RAT speleothem U/Th ages with the 2σ error bars are found at the top of the figure together with the speleothem dates from the Ejulve and Corchia caves. A) Precession curve with pink vertical bars highlighting minima precession (Laskar, 2004). B) RAT Mg/Ca record with periods of enhanced precipitation periods marked with horizontal black lines, which correspond to interglacial/interstadial marine isotopic stages (MIS). C) The $\delta^{18}O_{speleo}$ Corchia record (Drysdale *et al.*, 2004; Tzedakis *et al.*, 2018) and the $\delta^{13}C_{speleo}$ record from the Ejulve cave (Pérez-Mejias *et al.*, 2017). D) The Tenaghi Philippon arboreal pollen record from Greece (Tzedakis *et al.*, 2006), and E) the $\delta^{18}O_{speleo}$ record from Sanbao cave in China (Cheng *et al.*, 2012, 2016).

Similarly, long-term pollen records of the Mediterranean region show expansions of the arboreal vegetation coincident with precession minima because of warmer climate and greater water availability (Sánchez-Goñi et al., 2008; Tzedakis et al., 2006 and 2009; Wagner *et al.*, 2019). In particular, the arboreal pollen percentages in the Tenaghi Philippon pollen record from Greece, in the eastern Mediterranean region, show an extraordinary resemblance with the precipitation variability recorded in





the RAT Mg/Ca, where high arboreal pollen percentages are in line with enhanced precipitation (Fig. 3) (Tzedakis *et al*., 2003 and 2006). However, certain time discrepancies can be observed mostly during the end of the glacial MIS 10 and for the TIV and TIII.a. The Tenaghi Philippon age model relies on orbital tuning which also can account for some discrepancies with the absolute dated speleothem record. Aside from the age discrepancies during deglaciations, the magnitude of events and their structures are comparable with the variability recorded on RAT.


Precession also has a major effect on the Mediterranean oceanography, due to its ability to disrupt the evaporation-precipitation balance. Marine sediment records close to the Strait of Gibraltar recorded changes in the Mediterranean Outflow Water (MOW) strength, mainly driven by precession-controlled fluctuations in the Mediterranean hydrologic budget (Nichols *et al.,* 2020; Sierro *et al*., 2020), and marine sediments from the Balearic Sea documented carbonate cyclicities due to the control of the W

Mediterranean cyclogenetic mechanisms (Ochoa *et al.,* 2018). The precession control on the African monsoon is well known, which indirectly affected the eastern Mediterranean surface waters, through river runoff along the North African coast, changing drastically the $\delta^{18}O_{sw}$ composition and allowing sapropel formations (Rohling *et al.,* 2014; Grant *et al*., 2016). Several studies highlight that the timing of enhanced summer monsoon corresponds with maximum Mediterranean autumn/winter storm track precipitation (Kutzbach et al., 2014; Toucanne et al., 2015; Wagner *et al*., 2019). The Mg/Ca-RAT record also

reveals this concordance, where western Mediterranean enhanced precipitation, during interglacial/interstadial periods of precession minima, is consistently in line with higher summer monsoon intensities according to the Chinese speleothem records from Sanbao cave (Fig. 3) (Cheng *et al*., 2016).

**3.2 Onset of glacial terminations**

The newly obtained RAT records provide new constraints on the absolute age for glacial terminations TIV, TIII, and TIII.a

based on the identification of meltwater release reflected in negative $\delta^{18}O$ excursions (fig. 4A). Accordingly, the onset of TIV shows a first $\delta^{18}O$ depletion at 351±5 ka, followed by a larger depletion at 343±3 ka which likely reflecting the maximum rate of ice melting. This is consistent with a study based on volcanic interbed deposits of near-coastal aggradational successions in Italy (fig. 4A horizontal purple bars), which provided independent ages for sea level rise by means of $^{40}Ar/^{39}Ar$ absolute dating (Marra *et al*., 2016 and 2019). The onset of the TIII in the RAT speleothem is also in concordance with the sea-level rise

observed in Italy, and the principal $\delta^{18}O$ negative anomaly occurred at 245±2 ka. This anomaly is comparable in magnitude to the TIV freshening. The TIII.a deglaciation is marked by the negative $\delta^{18}O$ anomaly centered at 227±1 ka (Fig. 4). This characteristic $\delta^{18}O$ feature of deglaciations is also recognized in another Balearic speleothem (VALL2) covering the TV, although the high age uncertainties associated to the chronological limit of the U-Th technique preclude a precise date for the onset of this termination (Appendix D and Fig. D1).








**Figure 4: Comparison between glacial termination TIV (right panel) and glacial terminations TIII and TIII.a (left panel). On the top, the RAT U/Th ages with the 2σ error bars. The vertical blue bars highlight the duration of glacial terminations according to the negative δ¹⁸O structures of the RAT record, while the internal and thinner dark blue bars emphasize the maximum rate of freshenings discussed in the main text. A) The $\delta^{18}O_{speleo}$ record from RAT speleothem. The purple horizontal lines denote periods of sea-level rise according to Marra et al., 2016 and 2019. B) Orbital parameters (Laskar, 2004) and the caloric summer insolation**





**curve with maximums and minimums highlighted with red triangles (Tzedakis *et al*., 2006). C) The LR04 stack of δ¹⁸O benthic foraminifera (Lisiecki and Raymo, 2005) and the N. Atlantic IRD record of the ODP 983 site (Barker *et al*., 2015, 2019, 2021). D) The δ¹⁸O_{speleo} record of the Sanbao cave in China (Cheng *et al*., 2012, 2016). E) δ¹⁸O_{planktic} record from both the ODP 977 site and the MD01-2444 sediment core plotted using the speleo-tuned marine chronologies. The original chronology of the MD01-2444 core is in light green (Hodell *et al*., 2013).**

The peaks of maximum freshening for the three studied glacial terminations in RAT occurred during falling precession index and therefore consistent with the beginning of the summer insolation increases (Fig. 4B). In contrast, the obliquity phase is very variable among the three terminations. The TIV and TIII.a occurs during obliquity increase, while this parameter was decreasing through TIII. Hence, falling precession is the common orbital feature along the studied terminations, supporting the idea that precession appears more important than obliquity in predicting the onset of Late Pleistocene glacial terminations
(Hobart *et al*., 2023).

Tzedakis *et al*., 2017 proposed that the energy needed to force the glacial terminations also depends on the length and the intensity of the previous glacial period. The evaluation of the glacial conditions also reveals differences among the three studied terminations. Robust glacial conditions properly preconditioned the continental ice sheet volume before the TIV deglaciation.
Lower precession amplitude, together with minimum obliquity through the full glacial conditions of the MIS 10, led to less caloric summer insolation before the TIV onset, strengthening the glacial maximum conditions. The caloric summer insolation is controlled by the combination of both, precession, and obliquity, and represents the amount of energy integrated over the caloric summer half of the year, certainly playing a causal role in the achievement of interglacial acmes (Milankovic 1941; Tzedakis *et al*., 2006). But it should also influence glacial attributes. The MIS 10 culminated with a maximum in the ice-sheet
extent according to the benthic δ¹⁸O record from the LR04 stack (Lisiecki and Raymo, 2005), but also observable in the more extreme positive glacial δ¹⁸O in the δ¹⁸O RAT record. Furthermore, IRDs were released continuously in the North Atlantic at the end of the MIS 10, contributing to freshwater inputs (Fig. 4C) (Barker *et al*., 2015, 2019 and 2021). These intense predeglacial conditions facilitated the early freshening phase at the onset of TIV and, may have reinforced the duration of the subsequent pronounced freshening and the relaxation time observed in the δ¹⁸O RAT record. The maximum glacial conditions
with noticeable IRD release did not present such a prominent ice volume maximum at the end of the MIS 8, neither in the benthic δ¹⁸O nor in the RAT records. This glacial situation favored the shorter and less noticeable freshening of the TIII (Fig. 4C). The δ¹⁸O RAT shows a similar freshening structure between TIII and TIII.a, despite TIII.a being an exceptional deglaciation because it does not proceed from full glacial conditions. Short cold stadial conditions, with less IRD release and accumulated ice volume, allowed anyway a strong response of the ice-sheet dynamics to the insolation rise probably due to
the pronounced orbital changes through this deglaciation. Interestingly, the meltwater release of TIII.a culminated significantly before the summer caloric insolation maxima, reinforcing the exceptional character of this extra termination (Fig.4C), while TIV and TIII are closer to their respective summer caloric insolation maxima. The extent of the studied freshenings is in line with AMOC weakening according to the Sanbao Chinese speleothems (Fig.4D).





A comparison among the last five glacial terminations is presented in Figure 5. The studied terminations (TIV, TIII, and TIII.a) are directly compared with terminations TII and TI recorded in the $\delta^{18}O$ records of Cantabrian speleothems, from north Iberia Peninsula caves (Stoll *et al.*, 2015 and 2022). The TII and TI also occurred in line with falling precession index. In the right panel, all the records were centered at the first and most prominent $\delta^{18}O$ negative anomaly structure in order to compare their duration. The same exercise was performed with the isotopic measurements to observe the freshening magnitudes. This comparison underlines that TIII and TIII.a were those with less freshening magnitude while TIV was prominent. Unlike the benthic $\delta^{18}O$ records which feature a freshening and plateau at interglacial values, in the speleothem records the peak freshening marked by the negative $\delta^{18}O$ relaxes back to intermediate values. During TII, the period of most extreme negative $\delta^{18}O$ in North Iberia speleothems indicates fresher surface ocean $\delta^{18}O_{sw}$, highlighting the concentration of meltwater in the North Atlantic, and the relaxation has been interpreted to reflect the mixing of this meltwater anomaly throughout the global ocean as the rate of meltwater addition slows and the rate of overturning circulation strengthens (Stoll *et al.*, 2022). The RAT record suggests differences in the time required from the deglacial onset until the relaxation of these negative $\delta^{18}O$ anomalies, from very rapid (~5 kyr) during TIII.a, similar to TI, intermediate (~10 kyr) for TIII similar to TII, and much slower (~20 kyr) for TIV (Fig. 5).











**Figure 5: Comparison of the $\delta^{18}O_{speleo}$ records from the RAT speleothem and the North Iberian speleothems (Stoll *et al.*, 2022) along the last five glacial terminations. In the left panel, the speleothem records are plotted against the obliquity and precession parameters (Laskar, 2004). The onset of each interglacial acme is highlighted with a vertical pink line according to the Sanbao speleothem record (Cheng *et al.*, 2012, 2016), which considered the recovery of the Eastern Asian Summer Monsoon intensities as the end of the deglaciation. The right panel shows the duration and the magnitude of the $\delta^{18}O$ freshening structures using differential values in both the X and Y axes. The zero value corresponds to the first and most negative $\delta^{18}O$ value of the main freshening structure of each termination. The black arrow underlines the freshening relaxation of the negative isotopic anomaly after the maximum freshening rate. The maximum summer caloric insolation after each glacial termination is also highlighted with a red circle. It is notable how different the positions of these maximums are with respect to the freshening structure and the interglacial acme among the compared terminations.**

## 3.2 Implications for Marine Chronologies

The RAT speleothem provides absolute chronologies for the deglacial time sequences in the western Mediterranean, which can be transferred to marine sequences that have proxies also sensitive to deglacial freshening. This is the case of the $\delta^{18}O_{planktic}$ data measured in *G. bulloides* from the ODP 977 site in the Alboran Sea, where their combined use with Mg/Ca derived temperatures has provided $\delta^{18}O_{sw}$ reconstructions that prove the arrival of north Atlantic freshening associates to the major melting phases of mostly the Fennoscandian ice sheets during TV, TIII.a, TII and TI (Azibeiro *et al.,* 2021; Gonzalez-Mora *et al.,* 2008; Jiménez-Amat and Zahn 2015; Torner *et al.*, 2019; Stoll et al., 2022). These studies conducted at the ODP 977 site indicate that the timing and magnitude of the main negative $\delta^{18}O_{sw}$ shift are synchronous with $\delta^{18}O_{planktic}$ changes at glacial terminations, suggesting that the freshwater signal may have a dominant influence on the $\delta^{18}O_{planktic}$ variability (Appendix Fig. E1). Moreover, the amplitude between the deglacial anomalies of the $\delta^{18}O_{planktic}$ record in the ODP 977 and the $\delta^{18}O$ RAT records are similar for the three studied terminations. Hence, the new $\delta^{18}O_{planktic}$ data for TIV, TIII, and TIII.a provide an exceptional base to tune the millennial-scale features of the marine record to our dated $\delta^{18}O$ RAT record, achieving an orbitally independent chronology. Therefore, the light $\delta^{18}O_{planktic}$ structures, have been used to align the chronology with the $\delta^{18}O$ negative anomalies of RAT during the glacial terminations. The resulting ODP 977 age model is shown in Appendix Figure F1, where the tie points used by the alignment can be observed. Freshening due to ice-melting is notable in the $\delta^{18}O_{planktic}$ ODP 977 signature and the resemblance with the $\delta^{18}O$ RAT record is remarkable during glacial terminations (Fig.4E). This new ODP 977 chronology is further tuned to records from the Atlantic Iberian Margin (MD01-2444), also containing deglacial light $\delta^{18}O_{planktic}$ anomalies (Fig.4E) (Tzedakis *et al.*, 2018; Hodell *et al.*, 2013). Several tie points have also been used to tune the MD01-2444 in the base of the most relevant structures (Appendix Fig. F1). All the correlations among the tuned records were estimated by linear interpolation using the Analyseries Software package (Paillard *et al.*, 1996). This exercise reveals significant time discrepancies, particularly for TIV and TIII.a, and exposes that marine chronologies remain unconstrained mostly for glacial termination onsets. Supporting the idea that millennial-scale variability could strongly influence climate variation along deglaciations (Barker et al., 2021) and highlighting that the distinctive characteristic of each glacial termination should be seriously considered to tune the marine chronologies. This study exposes the power of the RAT speleothem as an excellent example to evaluate the time sequences of TIV and encourages the revision of the marine chronologies to clarify glacial termination dynamics independently of the orbital bond.



## 4 Conclusions

Geochemical proxies in the speleothem RAT reveal alternating dry glacial conditions and wet interglacial conditions in the western Mediterranean region from MIS 11 to MIS 7. Deglaciations were systematically initiated with major ocean freshening events in the North Atlantic, recorded as negative $\delta^{18}O$ anomalies in the speleothem, and in parallel with extreme arid
conditions in the western Mediterranean region according to the Mg/Ca. This study provides further evidence that falling precession was the major orbital trigger for glacial terminations and constrains the deglacial time sequences of each studied glacial termination. Despite the common features, the three deglaciations present differences in terms of magnitude, intensity, and duration. The dissimilarities along the preceding glacial conditions also contributed to these distinctive features. TIV was slower, longer, and more intense than TIII and TIII.a. The RAT record provides an absolute chronology for both the periods
of intense meltwater release initiating glacial terminations and the maximum freshening of each deglaciation: TIV onset at 351±5 ka BP and maximum freshening peak at 343±3 ka, the TIII onset at 251±3 ka and the maximum freshening peak at 245±2 ka, and finally the TIII.a onset at 229±2 ka BP and the maximum freshening peak at 227±1 ka. The new absolute chronology suggests that revision is needed to the most common marine chronologies mostly for TIV, which are based on orbital tuning. This discrepancy, and the role of millennial-scale variability associated with the onset of glacial terminations,
suggests that a standard orbital tuning of marine sequences will often not yield a robust chronology. The RAT speleothem provides for the very first time the possibility to validate and improve the accuracy of North Atlantic and Western Mediterranean marine chronologies by independent absolute dating.

## 5 Appendices

### Appendix A) U/Th laboratory procedure

Speleothem carbonate sample powders were dissolved in 7N $HNO_3$ and spiked with an in-house solution with known $^{233}U$, $^{236}U$, and $^{229}Th$ concentrations. After that, a few drops of concentrated $HClO_4$ were added to attack and remove organic matter. After one hour of refluxing, the samples were completely dried and re-dissolved in 2N HCl. The U and Th were isolated from Ca and several trace elements by means of an iron co-precipitation. Then, 1-2 drops of $FeCl_2$ were added, and several $NH_4OH$ drops were added later to induce the Fe precipitation together with U and Th. Afterward, samples were centrifuged and rinsed
with super-clean water three times while the overlying liquid was removed. In order to separate Fe from U and Th, iron samples were dissolved and loaded in anion exchange resin columns. While Fe was removed with 7N $HNO_3$, Th was collected with 6N HCl and separated from U, which was recovered ultimately with super-clean water. Each pair of samples were dried and dissolved in a low-concentrated nitric acid and analyzed by means of multi-collector inductively coupled plasma mass spectrometry at the University of Minnesota and the Xi'an Jiaotong University. Results and dating errors of the RAT and
VALL 2 speleothems are found in Table A1.



$^{230}$Th dating results. The error is 2σ error.

| Sample number | Depth (cm) | Sample Label | $^{238}$U (ppb) | $^{232}$Th (ppt) | $^{230}$Th / $^{232}$Th (atomic x10⁻⁶) | δ$^{234}$U* (measured) | $^{230}$Th / $^{238}$U (activity) | $^{230}$Th Age (yr) (uncorrected) | $^{230}$Th Age (yr) (corrected) | δ$^{234}$U$_{initial}$** (corrected) | $^{230}$Th Age (yr BP)*** (corrected) |
|---|---|---|---|---|---|---|---|---|---|---|---|
| 1 | 0.75 | RAT_A | 186.0 ±0.2 | 1697 ±34 | 3951 ±79 | 1249.1 ±2.5 | 2.1858 ±0.0039 | 210673 ±1087 | 210591 ±1088 | 2263 ±8 | 210525 ±1088 |
| 2 | 7.25 | RAT_A.1 | 125.6 ±0.1 | 1179 ±24 | 3947 ±79 | 1246.2 ±2.2 | 2.2458 ±0.0033 | 226066 ±1054 | 225984 ±1055 | 2358 ±8 | 225918 ±1055 |
| 3 | 7.25 | RAT-2.75 | 177.3 ±0.1 | 106 ±2 | 62119 ±1434 | 1257.5 ±1.9 | 2.2584 ±0.0033 | 226101 ±993 | 226095 ±993 | 2380 ±8 | 226024 ±993 |
| 4 | 8.5 | RAT_B | 118.7 ±0.1 | 645 ±13 | 6804 ±137 | 1235.1 ±2.7 | 2.2415 ±0.0044 | 228149 ±1397 | 228102 ±1397 | 2351 ±11 | 228036 ±1397 |
| 5 | 9.75 | RAT_B.1 | 83.8 ±0.1 | 1188 ±24 | 2542 ±51 | 1200.3 ±2.3 | 2.1862 ±0.0052 | 223677 ±1510 | 223550 ±1511 | 2256 ±11 | 223484 ±1511 |
| 6 | 13.5 | RAT_B.2 | 85.7 ±0.1 | 1269 ±25 | 2389 ±48 | 1135.8 ±2.1 | 2.1465 ±0.0037 | 232313 ±1265 | 232178 ±1267 | 2187 ±9 | 232112 ±1267 |
| 7 | 14 | RAT 2 | 85.5 ±0.1 | 284 ±6 | 10808 ±221 | 1156.0 ±1.9 | 2.1804 ±0.0031 | 235607 ±1093 | 235576 ±1093 | 2247 ±8 | 235505 ±1093 |
| 8 | 15 | RAT c | 54.6 ±0.1 | 584 ±12 | 3312.0 ±66.6 | 1120.2 ±2.1 | 2.1475 ±0.0035 | 237779 ±1277 | 237680 ±1278 | 2191 ±8.9 | 237616 ±1278 |
| 9 | 15.25 | RAT 3.25 | 70.9 ±0.1 | 1169 ±23 | 2108 ±42 | 1085.7 ±1.6 | 2.1074 ±0.0036 | 237367 ±1258 | 237213 ±1261 | 2121 ±8 | 237142 ±1261 |
| 10 | 15.5 | RAT_C.1 | 76.1 ±0.1 | 4946 ±99 | 519 ±10 | 1009.5 ±2.0 | 2.0490 ±0.0038 | 246517 ±1518 | 245888 ±1574 | 2021 ±10 | 245822 ±1574 |
| 11 | 16 | RAT_D | 75.3 ±0.1 | 2113 ±42 | 1159 ±23 | 925.9 ±2.0 | 1.9725 ±0.0033 | 253671 ±1568 | 253388 ±1578 | 1893 ±9 | 253326 ±1578 |
| 12 | 16.5 | RAT 4.5 | 56.9 ±0.0 | 457 ±9 | 4187 ±85 | 926.5 ±1.7 | 2.0405 ±0.0039 | 282375 ±2087 | 282297 ±2087 | 2055 ±13 | 282226 ±2087 |
| 13 | 17 | RAT 5 | 67.7 ±0.1 | 75 ±2 | 29556 ±732 | 887.3 ±1.7 | 1.9982 ±0.0034 | 284836 ±1946 | 284824 ±1946 | 1982 ±12 | 284753 ±1946 |
| 14 | 17.7 | RAT 5.7 | 71.3 ±0.1 | 110 ±2 | 20978 ±466 | 859.2 ±1.6 | 1.9700 ±0.0036 | 287778 ±2109 | 287761 ±2109 | 1935 ±12 | 287690 ±2109 |
| 15 | 18.25 | RAT_E | 78.5 ±0.1 | 654 ±13 | 3843 ±77 | 834.4 ±2.0 | 1.9399 ±0.0038 | 287767 ±2405 | 287682 ±2404 | 1879 ±14 | 287620 ±2404 |
| 16 | 20 | RAT_E.1 | 90.0 ±0.1 | 445 ±9 | 6121 ±124 | 738.5 ±2.2 | 1.8340 ±0.0036 | 293865 ±2657 | 293810 ±2656 | 1692 ±14 | 293744 ±2656 |
| 17 | 23 | RAT_F | 94.2 ±0.1 | 657 ±13 | 4069 ±82 | 636.2 ±1.8 | 1.7211 ±0.0034 | 301956 ±2809 | 301874 ±2808 | 1491 ±13 | 301812 ±2808 |
| 18 | 24 | RAT 12 | 79.5 ±0.1 | 28 ±1 | 77872 ±3369 | 571.6 ±1.4 | 1.6549 ±0.0027 | 312266 ±2597 | 312260 ±2597 | 1380 ±13 | 312189 ±2597 |
| 19 | 26 | RAT_F.1 | 80.5 ±0.1 | 55 ±2 | 38662 ±1081 | 528.9 ±1.9 | 1.6075 ±0.0034 | 317242 ±3246 | 317232 ±3246 | 1295 ±13 | 317166 ±3246 |
| 20 | 28 | RAT_G | 96.1 ±0.1 | 230 ±5 | 10923 ±221 | 515.8 ±1.9 | 1.5882 ±0.0034 | 314689 ±3633 | 314657 ±3632 | 1253 ±14 | 314595 ±3632 |
| 21 | 28.75 | RAT_H | 87.6 ±0.1 | 686 ±14 | 3275 ±66 | 486.5 ±1.7 | 1.5556 ±0.0035 | 318254 ±3778 | 318148 ±3776 | 1194 ±13 | 318086 ±3776 |
| 22 | 29.5 | RAT_H.1 | 103.9 ±0.1 | 172 ±4 | 15358 ±318 | 476.0 ±2.0 | 1.5440 ±0.0034 | 319752 ±3689 | 319729 ±3689 | 1173 ±13 | 319663 ±3689 |
| 23 | 32 | RAT_H.2 | 63.9 ±0.1 | 144 ±3 | 11459 ±245 | 485.3 ±1.7 | 1.5627 ±0.0030 | 326523 ±3661 | 326492 ±3661 | 1219 ±13 | 326426 ±3661 |
| 24 | 32.75 | RAT_I | 89.9 ±0.1 | 153 ±3 | 15552 ±315 | 520.6 ±1.8 | 1.6073 ±0.0039 | 326426 ±4204 | 326404 ±4203 | 1308 ±16 | 326342 ±4203 |
| 25 | 34 | RAT_I.1 | 72.7 ±0.1 | 217 ±4 | 9021 ±186 | 535.3 ±1.9 | 1.6372 ±0.0033 | 337484 ±4138 | 337446 ±4137 | 1387 ±17 | 337380 ±4137 |
| 26 | 35 | RAT_J | 81.5 ±0.1 | 162 ±3 | 13893 ±281 | 573.2 ±1.8 | 1.6805 ±0.0036 | 332647 ±3944 | 332622 ±3943 | 1465 ±17 | 332560 ±3943 |
| 27 | 35.75 | RAT 23.75 | 81.9 ±0.1 | 100 ±2 | 22796 ±523 | 576.1 ±1.4 | 1.6865 ±0.0030 | 334834 ±3324 | 334818 ±3323 | 1482 ±14 | 334747 ±3323 |
| 28 | 36.6 | RAT 24.6 | 90.5 ±0.1 | 470 ±10 | 5671 ±115 | 655.1 ±1.6 | 1.7788 ±0.0027 | 335551 ±2982 | 335493 ±2981 | 1688 ±15 | 335422 ±2981 |
| 29 | 36.75 | RAT_J.1 | 80.7 ±0.1 | 195 ±4 | 11499 ±240 | 573.5 ±1.8 | 1.6833 ±0.0034 | 334990 ±3912 | 334960 ±3911 | 1476 ±17 | 334894 ±3911 |
| 30 | 37 | RAT 25 | 99.5 ±0.1 | 428 ±9 | 7265 ±147 | 731.9 ±1.5 | 1.8963 ±0.0030 | 344621 ±3138 | 344576 ±3137 | 1935 ±18 | 344505 ±3137 |
| 31 | 37.25 | RAT_J.2 | 85.1 ±0.1 | 572 ±12 | 5019 ±102 | 838.0 ±2.2 | 2.0452 ±0.0041 | 355336 ±4289 | 355275 ±4288 | 2284 ±28 | 355209 ±4288 |
| 32 | 37.5 | RAT 25.5 | 100.9 ±0.1 | 556 ±11 | 6141 ±124 | 845.7 ±1.6 | 2.0529 ±0.0035 | 353380 ±3159 | 353330 ±3158 | 2292 ±21 | 353259 ±3158 |
| 33 | 38.5 | RAT_J.3 | 122.2 ±0.1 | 113 ±2 | 35121 ±759 | 768.4 ±1.8 | 1.9654 ±0.0028 | 365001 ±3564 | 364990 ±3564 | 2152 ±22 | 364924 ±3564 |
| 34 | 39 | RAT 27 | 134.4 ±0.1 | 59 ±2 | 71303 ±1884 | 712.5 ±1.5 | 1.9010 ±0.0031 | 373880 ±4055 | 373874 ±4055 | 2047 ±24 | 373803 ±4055 |
| 35 | 40.1 | RAT_J.4 | 151.9 ±0.2 | 59 ±2 | 80965 ±2468 | 716.0 ±1.9 | 1.9038 ±0.0036 | 372020 ±4716 | 372014 ±4716 | 2046 ±28 | 371948 ±4716 |
| 36 | 45 | RAT_J.5 | 146.9 ±0.2 | 53 ±2 | 86817 ±2581 | 723.2 ±2.0 | 1.9102 ±0.0033 | 368865 ±4477 | 368860 ±4477 | 2048 ±27 | 368794 ±4477 |
| 37 | 47 | RAT_J.6 | 118.5 ±0.1 | 45 ±1 | 81924 ±2446 | 716.5 ±2.1 | 1.9001 ±0.0035 | 367292 ±4708 | 367287 ±4708 | 2020 ±28 | 367221 ±4708 |
| 38 | 48.5 | RAT 36.5 | 117.5 ±0.1 | 183 ±4 | 20120 ±425 | 712.1 ±1.6 | 1.9027 ±0.0044 | 376488 ±5495 | 376472 ±5494 | 2060 ±32 | 376401 ±5494 |
| 39 | 50 | RAT_J.7 | 92.8 ±0.1 | 52 ±2 | 55752 ±1711 | 709.6 ±2.5 | 1.9067 ±0.0049 | 384890 ±7090 | 384883 ±7090 | 2103 ±43 | 384817 ±7090 |
| 40 | 50.7 | RAT 38.7 | 70.3 ±0.0 | 41 ±1 | 53378 ±1862 | 709.7 ±1.9 | 1.9042 ±0.0037 | 381728 ±5194 | 381721 ±5194 | 2084 ±31 | 381650 ±5194 |
| 41 | 51.5 | RAT k | 86.0 ±0.1 | 258 ±5 | 10410 ±211 | 707.5 ±2.0 | 1.8948 ±0.0033 | 374408 ±4678 | 374378 ±4677 | 2035 ±27 | 374314 ±4677 |

$^{230}$Th dating results. The error is 2σ error.

| Sample number | Depth (cm) | Sample Number | $^{238}$U (ppb) | $^{232}$Th (ppt) | $^{230}$Th / $^{232}$Th (atomic x10⁻⁶) | δ$^{234}$U* (measured) | $^{230}$Th / $^{238}$U (activity) | $^{230}$Th Age (yr) (uncorrected) | $^{230}$Th Age (yr) (corrected) | δ$^{234}$U$_{initial}$** (corrected) | $^{230}$Th Age (yr BP)*** (corrected) |
|---|---|---|---|---|---|---|---|---|---|---|---|
| 1 | 0.2 | VALL2_0.2 | 1294.6 ±3.1 | 41 ±1 | 452066 ±15597 | 108.8 ±2.0 | 0.8697 ±0.0025 | 159998 ±1252 | 159997 ±1252 | 171 ±3 | 159931 ±1252 |
| 2 | 1.8 | VALL2_Top | 342.8 ±0.6 | 12 ±0 | 485542 ±11926 | 39.1 ±1.7 | 1.0199 ±0.0030 | 377308 ±11561 | 377304 ±11561 | 113 ±6 | 377242 ±11561 |
| 3 | 4.3 | VALL2_4.3 | 289.3 ±0.4 | 28 ±1 | 170650 ±5044 | 32.0 ±1.5 | 1.0152 ±0.0018 | 392841 ±9902 | 392835 ±9902 | 97 ±5 | 392769 ±9902 |
| 4 | 5.5 | VALL2_5.5 | 334.2 ±0.5 | 4 ±1 | 1295999 ±278065 | 28.8 ±1.7 | 1.0207 ±0.0017 | 436721 ±15916 | 436716 ±15915 | 99 ±7 | 436650 ±15915 |
| 5 | 7 | VALL2_7 | 315.9 ±0.4 | 23 ±1 | 230927 ±13011 | 31.1 ±1.5 | 1.0199 ±0.0021 | 416922 ±13354 | 416916 ±13353 | 101 ±6 | 416850 ±13353 |
| 6 | 7.6 | VALL2_7.6 | 284.5 ±0.4 | 10 ±1 | 503033 ±40961 | 35.3 ±1.7 | 1.0281 ±0.0018 | 429141 ±14893 | 429136 ±14893 | 118 ±8 | 429070 ±14893 |
| 7 | 8.1 | VALL2_8.1 | 372.4 ±0.6 | 19 ±1 | 334429 ±20523 | 40.4 ±1.8 | 1.0395 ±0.0020 | 454126 ±19373 | 454119 ±19372 | 145 ±10 | 454053 ±19372 |
| 8 | 8.6 | VALL2_8.6 | 246.9 ±0.3 | 36 ±1 | 118137 ±4037 | 39.7 ±1.5 | 1.0416 ±0.0020 | 474894 ±21176 | 474883 ±21175 | 151 ±11 | 474817 ±21175 |
| 9 | 9 | VALL2_9 | 770.8 ±1.8 | 10 ±1 | 1321617 ±104976 | 31.0 ±2.0 | 1.0321 ±0.0026 | 494581 ±34732 | 494572 ±34730 | 125 ±15 | 494506 ±34730 |
| 10 | 12 | VALL2_12 | 811.2 ±2.3 | 56 ±2 | 247280 ±6969 | 28.4 ±2.6 | 1.0284 ±0.0035 | 540121 ±73096 | 540107 ±73086 | 117 ±28 | 540041 ±73086 |
| 11 | 15.6 | VALL2_15.6 | 211.6 ±0.2 | 24 ±1 | 148538 ±6320 | 25.4 ±1.5 | 1.0246 ±0.0019 | 495996 ±26282 | 495984 ±26280 | 103 ±10 | 495918 ±26280 |
| 12 | 18 | VALL2_18 | 335.5 ±0.4 | 29 ±1 | 194361 ±7398 | 25.1 ±1.5 | 1.0260 ±0.0020 | 514267 ±32267 | 514255 ±32264 | 107 ±12 | 514189 ±32264 |
| 13 | 20.3 | VALL2_Bottom | 349 ±1 | 11 ±1 | 549205.7 ±32644.0 | 23.9 ±1.5 | 1.0251 ±0.0033 | 523624 ±48562 | 523612 ±48557 | 105 ±16 | 523550 ±48557 |

U decay constants: $\lambda_{238} = 1.55125 \times 10^{-10}$ (Jaffey et al., 1971) and $\lambda_{234} = 2.82206 \times 10^{-6}$ (Cheng et al., 2013). Th decay constant: $\lambda_{230} = 9.1705 \times 10^{-6}$ (Cheng et al., 2013).

*δ$^{234}$U = ([$^{234}$U/$^{238}$U]$_{activity}$ − 1)x1000. ** δ$^{234}$U$_{initial}$ was calculated based on $^{230}$Th age (T), i.e., δ$^{234}$U$_{initial}$ = δ$^{234}$U$_{measured}$ x e$^{\lambda_{234}xT}$.

Corrected $^{230}$Th ages assume the initial $^{230}$Th/$^{232}$Th atomic ratio of 4.4 ±2.2 x10⁻⁶. Those are the values for a material at secular equilibrium, with the bulk earth $^{232}$Th/$^{238}$U value of 3.8. The errors are arbitrarily assumed to be 50%.

***B.P. stands for "Before Present" where the "Present" is defined as the year 1950 A.D.

**Table A1: U/Th dates from RAT and VAll 2 speleothems**





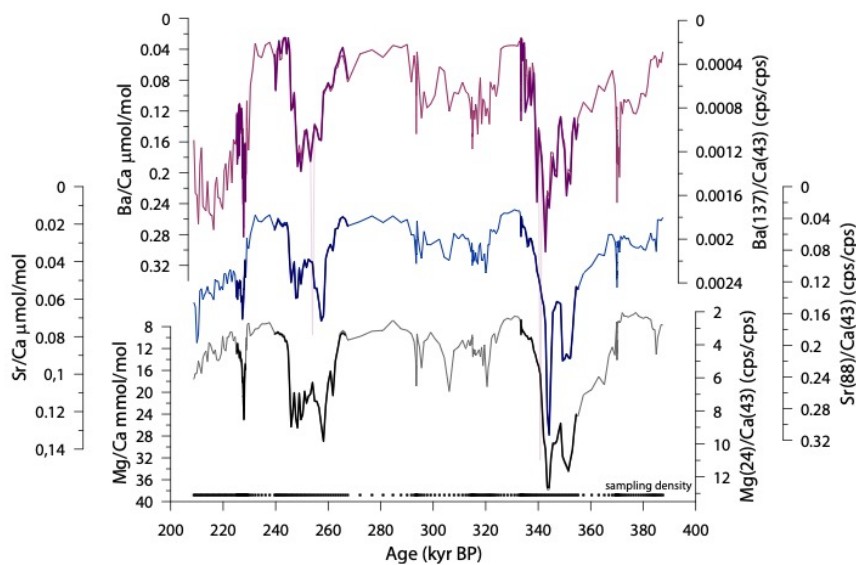


**Figure B1: RAT Mg/Ca ratios compared with Sr/Ca and Ba/Ca ratios in both, counts per second, right axes, and the calculated mmol/mol and µmol/mol values at left axes.**

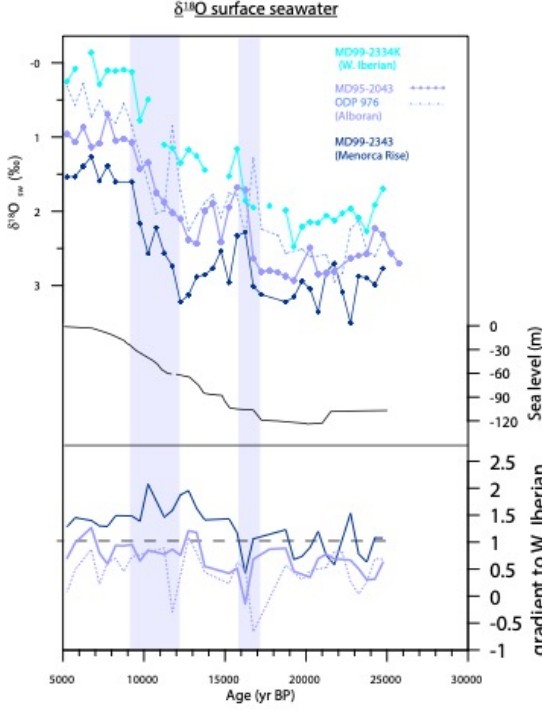

**Figure C1: Transmission of the seawater isotopic signal from the N. Atlantic to the Mediterranean Sea along the TI modified from**
**Stoll *et al.*, (2022). The $\delta^{18}O_{sw}$ values from paired Mg/Ca and $\delta^{18}O_{plank}$ decline along the deglaciation, with two well-recognized major freshening steps, one abrupt around 16 ka, and the other more gradual from about 12 to 10 ka (blue vertical bars). These freshenings were recorded synchronously and with a similar magnitude in different sediment records from three different locations. The core MD99-2334K from the western Iberian margin in the N. Atlantic (Skinner *et al.*, 2006), two sediment cores from the Alboran Sea in**





**the western Mediterranean (Català *et al.,* 2019), both close to the ODP 977 site. And the MD99-2343 core from the Balearic Sea**
**(Català *et al.,* 2019). Sea level rise from Lambeck *et al*., (2014). In the bottom panel, the gradient of the isotopic signal between the Alboran or the Balearic Sea against the W. Iberian record shows the characteristic enrichment of the seawater throughout the Mediterranean Sea, with probably higher evaporative processes in the Menorca rise leading to even heavier values in this basin.**

**Appendix D) Vallgornera speleothem record and the T-V**

*El Pas de Vallgornera* cave is in the south region of Mallorca Island (39° 22' 00'' N, 2° 52' 25'' E) (Fig. 1). This cave consists of more than 67 km of flooded conduits and chambers. Sea level reconstruction by means of overgrowths on speleothems has been previously studied in this cave (Dorale *et al.*, 2010). The VALL2 speleothem is a wide dark-brown speleothem of 22 cm long. The color at the bottom of the speleothem is light brown with a porous texture that towards the top becomes darker and

less porous. The growth axis presents several changes in the direction and different discontinuities represented by light layers. After the last main discontinuity, the speleothem presents two independent growths. The axis selected to accomplish the geochemical analysis has a 22 cm length and follows the most developed axis of the last two growths (Fig. C1). VALL2 spans from 530 to 345 ka BP, close to the U-series age limit. The chronology is provided by 12 radiometric dates although some of them are reversals. The age uncertainties are higher in the oldest part of the speleothem where the growth rate is higher and

coincides with a light-coloured area. The growth rate ranges between 0.02-1.2 cm/kyr. According to the age model, VALL 2 started to grow at the end of MIS 13 and stopped at the end of MIS 11. It presents a remarkable glacial structure (MIS 12: 510-455 ka BP) with heavier $\delta^{18}O$, $\delta^{13}C$, and higher Mg/Ca ratios, coincident with three major changes in the growth axis direction and color laminations. After the glacial maximum, around 455 ka BP, the three geochemical records agree in a transition toward low/light values that correspond to Termination V, with similar structures to those discussed in the main text for the

RAT speleothem. The overall variation is 3.4 ‰, 3.8 ‰, and 15.7 mmol/mol for $\delta^{18}O$, $\delta^{13}C$, and Mg/Ca ratio respectively.



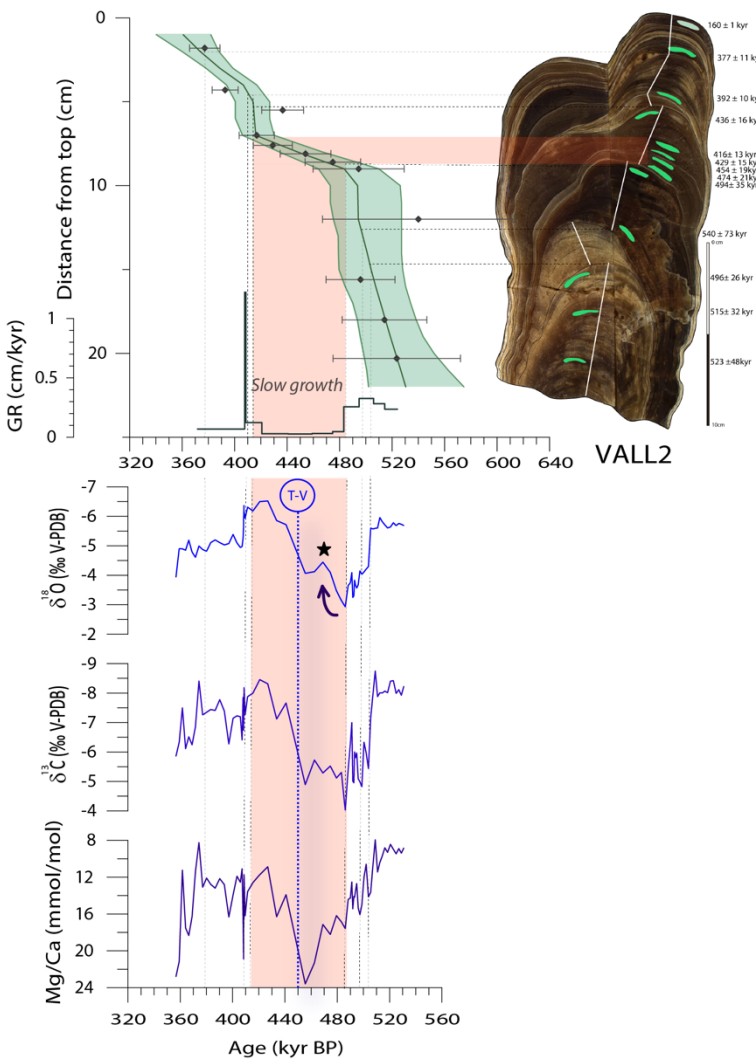

**Figure D1: VALL 2 speleothem age model acquired by StalAge (dark blue line) and the original U/Th dates (dark blue dots), plotted with the associated errors (error bars and green area). The VALL2 growth rates are shown in dark green. On the right, there is the speleothem image with the position of the sampled radiometric dates used for the age model and the growth axis path used for the geochemical analyses (white line). In the lower panel the δ18O, δ13C, and Mg/Ca results with reversal axes. The pink bar highlights a slow growth rate and the dashed grey line points to major color and growth axis direction changes of the speleothem.**



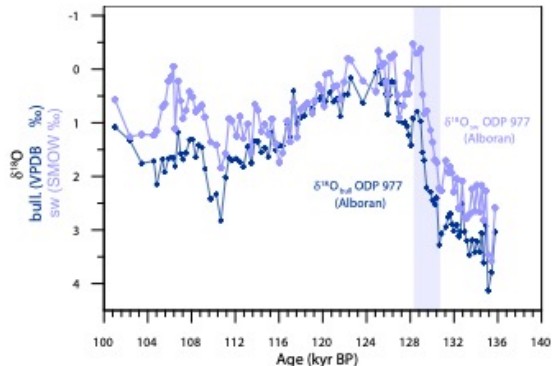

**Figure E1: Comparison between the *G. bulloides* δ¹⁸O$_{plank}$ of the ODP 977 site and the δ¹⁸O$_{sw}$ signal, obtained by removing the temperature effect using the derived SST-Mg/Ca also measured in *G. bulloides* of the same sediment core along glacial termination II (Torner *et al*., 2019). The maximum rate of freshening along deglaciation coincides in both records as rapid negative δ¹⁸O anomalies and is highlighted by a blue bar.**

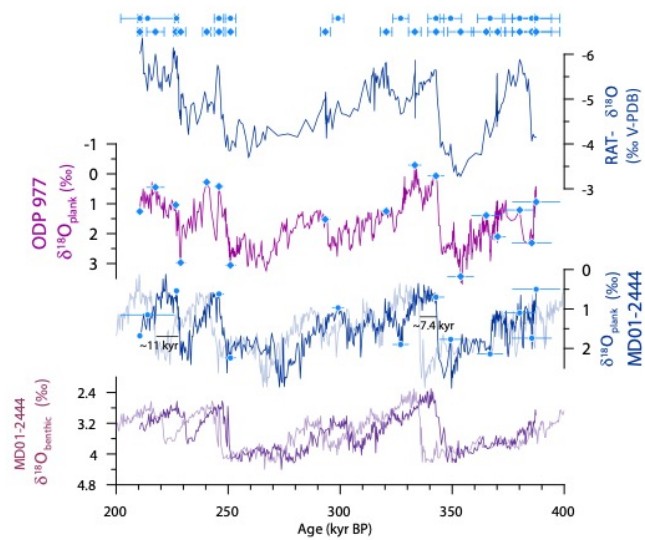

**Figure F1: Marine sediment tuning strategy using the RAT age model. At the top, blue circles and diamonds indicate the tie points used for chronological alignment with the speleothem errors associated according to the RAT δ¹⁸O record. The ODP 977 δ¹⁸O record on the RAT speleothem chronology is in the middle. At the bottom, the MD01-2444 planktic and benthic δ¹⁸O records on the RAT speleothem chronology and with the original chronology in light color (Tzedakis *et al*., 2018).**



## 6 Data availability

Upon acceptance, data will be presented for inclusion in the subsequent version of SISAL.

## 7 Author contributions

JT and IC designed and conceptualized the study. JT conducted measurements and analyzed the results. AM and HS provided scientific guidance. HC and RLE provided laboratory facilities and funding for U/Th measurements. FJS and JOG provided resources and interpretation of marine core ODP 977. JT was responsible for writing the original draft, while all authors contributed to the discussion and interpretations of the final manuscript.

## 8 Competing interests

The authors declare that they have no conflict of interest

## 9 Acknowledgements

This research was financially supported by the OPERA project (CTM2013-48639-C2-1-R), the CHIMERA project (CTM2016-75411-R), and the TRANSMOW project (PID2019-105523RB-I00) of the Ministerio de Ciencia e Innovación. Also has received funding from the European Research Council (ERC) Consolidation Grant (No. 683237-TIMED). JT and IC acknowledge the financial support by a Catalan Government Grups de Recerca Consolidats grant (2021 SGR 01195). The authors are grateful to M. Romero and J. Perona from CCiTUB for laboratory support. Thanks to M. Cerdà for his collaboration with the artworks of the maps. IC thanks the ICREA- Academia programme from the Generalitat de Catalunya.

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
