# Peer review of "New age constraints for glacial terminations IV, III, and III.a based on Western Mediterranean speleothem records"

_Climate of the Past, 2024_

## Author Comment (AC2)

**RC2**: Anonymous Referee #2, 27 Sep 2024

Torner and co-authors present a new speleothem record from Iberia that covers glacial terminations IV, III and III.a. Because of the exceptionally precise dating of the speleothem, this record provides new age constraints on the timing of glacial terminations in the Western Mediterranean region. These geochemical records provide information on termination dynamics in response to forcings, as well as their duration and the respective regional climatic conditions. While the last two glacial terminations are increasingly well constrained through absolute chronologies, much less is known about previous deglaciations, and this record fills an important gap in our understanding of late Quaternary deglaciation dynamics. Thus, it addresses a very relevant scientific question that is well within the scope of Climate of the Past. The methods are state of the art and well carried out, and the discussion is detailed and not overstating the significance of the dataset. This is a very well written and well organised manuscript, and I have only minor comments that hopefully will make it more accessible to the readers, when published.

We would like to thank the reviewer for their valuable comments on the relevance of our study. We agree with most of the suggestions and will incorporate them to improve the quality of the manuscript. We have addressed the comments below.

Minor comments:
- As far as I understand, this study includes new data from two speleothems, RAT and VALL2, and one marine foraminifera record. However, the manuscript only focuses on the RAT record, barely mentioning the other two datasets. I think it makes perfect sense for the study to focus on speleothem RAT, as the most complete and best dated record, but I think the manuscript would be clearer if the VALL2 and foraminifera records, their establishment, and provenance were described together with RAT in the methods, as well as mentioned in abstract and introduction (where the RAT record is mentioned). This would strengthen the manuscript as it would show that the results are based on more than a single speleothem record.

As you mentioned, our primary focus is on the RAT speleothem, as it is the most significant archive obtained in this study. However, we recognize that omitting the other two records in the abstract and introduction sections could lead to confusion. To reinforce the robustness of the manuscript, we will mention the sediment core in both the abstract and the introduction sections. We agree that including this information is essential for a comprehensive understanding of our approach in transferring speleothem chronologies to marine records.

Nevertheless, we believe that the second speleothem, VALL 2, should not be directly mentioned in the abstract, as its results are not of substantial relevance to the manuscript. This speleothem was mainly analyzed due to its interesting geochemical signal corresponding to glacial termination V, but its age model does not have sufficient robustness to evaluate the timing of events during this termination or its relationship with orbital parameters. For this reason, we first relegated it to the Appendix. However, we agree that including this information in the

Introduction and Methods will enhance the clarity of the manuscript (see other responses below).

- New Abstract with changes:

The full understanding of climate feedbacks responsible for the amplification of deglaciations requires robust chronologies for these climate transitions, but, in the case of marine records, radiocarbon chronologies are only possible for the last glacial termination. Although the assumed relationships between the marine isotopic record and the orbital parameters provide a first order chronology for glacial terminations, an independent chronological control allows the relationships between orbital forcing and the climate response to be evaluated over multiple previous terminations. To assess this, we present new geochemical records from the western Mediterranean, including speleothems and marine sediment cores. The most notable speleothem, identified as RAT, established a new long terrestrial climate record for this region, spanning Marine Isotope Stages 11 to 7. Its absolute U/Th dates provide an exceptional chronology for the glacial terminations IV, III, and III.a. The onset of these three glacial terminations was marked by rapid $\delta^{18}O$ depletions, reflecting ocean freshening by ice melting, thus providing an excellent tie point for regional marine records also sensitive to such freshening. This is exemplified by new $\delta^{18}O$ data of the ODP 977 site from the Alboran Sea, where the speleothem chronology was employed to adjust its age model. The new chronologies reveal an earlier onset of the deglacial melting for the TIV and TIII.a in contrast to the generally accepted marine chronologies and indicate that the duration of these deglaciations was variable, with TIV particularly longer (~20 kyr). This study also supports that the onset of deglacial melting always occurred during declining precession index while a nonunique relation occurred with the obliquity parameter.

- line 30: use "northern" instead of north hemisphere

Thank you. We will correct that.

- line 67 and following: I find this sentence is worded a bit strangely, as it juxtaposes records with a specific climatic interpretation (Asian monsoon) to those from a region (America). Maybe consider adding the key climate patterns for American records as well, I assume they were left out because the references are indicating both North and South American records.

We mention these records in this sentence just to highlight that there are not too many records that continuously span several deglaciations, although they are from distant regions. European and Mediterranean records are introduced later. We chose to focus solely on explaining the Asian monsoon interpretation, as it will be referenced throughout the discussion.

We changed this paragraph as follows:

Speleothems with a relatively unambiguous climatic interpretation of their proxies spanning continuously several glacial-interglacial cycles are restricted to a few studies. For instance, those focused on America that support orbital parameters as the predominant driver of glacial terminations, although limited to the two most recent cycles (Cruz *et al.*, 2005, 2009; Moesley *et al.*, 2016). A notable example is the case of Chinese speleothems that provided a great paleoclimatic assembled record of the Asian monsoon lasting 640 kyr and encompassing various deglaciations (Yuan *et al.*, 2004; Cheng *et al.*, 2006, 2009, 2016). First insights into the existence of the extra glacial termination TIII.a were previously reported in these Chinese speleothem records, which argue that this termination shows patterns of events equivalent to other glacial terminations despite being amid MIS 7 (Cheng *et al.*, 2009). Superimposed on a strong precessional…

- line 83 and following: following my first comment above, I would mention the foram and the VALL2 records here as well.

Yes, we agree. However, we will mention this information in the next paragraph when we introduce this study material in line 87 and the following:

i.e line 87: This study presents two new speleothem records from the Balearic Islands located in the western Mediterranean region (Fig. 1). The VALL 2 speleothem was recovered from Mallorca Island and encompassed the TV. The RAT speleothem recovered from Minorca Island encompasses two glacial-interglacial cycles, providing absolute chronologies for the glacial terminations TIV, TIII, and TIII.a. A geochemical multiproxy approach has been applied in both speleothems to investigate climate variability with a special focus on their glacial terminations. The discussion will be mostly focused on the RAT speleothem, which offers high-precision absolute dates and a unique continuous record not provided by any existing stalagmite archive within this region. The studied deglaciations present different intensities and durations and open an opportunity to investigate the relationship between orbital parameters and paleoclimatic records. We apply the RAT chronology to newly acquired planktic isotope measurements from the ODP 977 site in the Alboran Sea, highlighting the potential of this speleothem record as a referential base for marine chronologies. This approach aims to avoid the circularity of trying to understand the drivers and thresholds of climate variability using marine chronologies depending on orbital tuning.

- Material and methods: I would start this section with the description of both caves (Murada and El Pas de Vallgornera) and their respective speleothems, followed by the marine sediment core data. Then move on to the geochemical method descriptions. This would probably also help to streamline the manuscript a bit, since a lot of the methods are the same.

Following the suggestion made by the two referees, we will change both the structure and title of Section 2.1 for 'Cave Settings and Regional Hydroclimate'.

The section will begin with a description of both caves, incorporating the information from Es Pas de Vallgornera cave currently included in the Appendix, as you recommended. We will then continue with the climatic and hydrological context already included in this section, going from a local perspective of the Balearic Islands to the broader western Mediterranean region. We also agree to include the geochemical analyses of both speleothems within the same Method section to streamline the manuscript.

However, we would prefer to keep the sediment core information, including the location, description, published age model, and geochemical analysis of foraminifera performed in this study, all together in a separate section. We will remove this content from Subsection 2.2, 'Geochemical Analyses', and create a new one, Subsection 2.3, titled '$\delta^{18}O$ Analyses in Planktic Foraminifera from ODP 977 site'. We believe that this reorganization will offer to the sediment core section a more distinct identity.

- line 149: "reversed axis" instead of "reversal axis"
Thank you. We will correct that.

- line 192 and following: I would join the two sentences ("According to the depth-age model…, while the warm marine isotopic stages…"). Also I think it's confusing to use different terms to describe opposing isotopic trends (here "high" and "light"). Please stick with one.

We have chosen the terms low/high instead of light/heavy for speleothem measurements. Nevertheless, we would utilize the terms 'more negative/more positive' in reference to planktic foraminifera isotopes as they constitute standard terminology within this discipline.

Furthermore, we have adopted the suggestion to combine both sentences, but this paragraph will change a bit due to the incorporation of the VALL 2 information in the main text. We also add here a statement explaining that we focused the discussion on the RAT speleothem.

 Section 3.1, now titled "*The climate signal in speleothems*" will start as follows:

Line 190: Mg/Ca ratios and stable isotopes in both speleothems reveal climate variability in the western Mediterranean region. According to the obtained depth-age models, the VALL 2 spans from 530 to 345 ka BP (MIS 13-11), encompassing the glacial termination V. While RAT spans from 387 to 209 ka BP (MIS 11-7), covering continuously an exceptional long-time-interval (178 kyr) that includes three glacial terminations (TIV, TIII, and TIII.a). Glacial conditions are characterized by slower growth rates and high $\delta^{18}O$, $\delta^{13}C$, and Mg/Ca, while the warm marine isotopic stages feature low isotopes and Mg/Ca ratios (Fig. 2 and C1). The RAT record presents an overall variation of 3.2‰, 6.3‰ and 24.6 mmol/mol for $\delta^{18}O$, $\delta^{13}C$, and Mg/Ca, respectively. And the overall variation is 3.4 ‰, 3.8 ‰, and 15.7 mmol/mol in the VALL 2 speleothem. Given the insufficient robustness of the VALL 2 age model for

the precise evaluation of climatic events, the discussion is primarily focused on the RAT results.

- line 196 and following: in the discussion on PCP mechanisms, I'm missing the potential influence of cave ventilation, which can drive PCP as well as drip rates. Is this not a factor at the studied caves or why was this mechanism omitted?

Yes, most of the known Balearic caves have ventilation related to changes in the atmospheric temperature outside the cave between seasons. During winter, the air is ventilated, and the cave typically presents lower levels of $CO_2$. We assume that this process could also be favored during colder periods due to a reduction in the respiration of the plant roots reaching the ceiling of the caves. For that reason, we adapted the text to briefly introduce this information as you suggested:

Line 196: The Mg/Ca ratio in speleothems is sensitive to variations in the precipitation/evaporation balance, reflecting changes in the amount of water infiltrating into the karst system. Frequently, arid conditions, characterized by reduced water infiltration, may increase the groundwater residence time in the karst and even reduce drip rates in the caves (Fairchild *et al.,* 2000). This scenario provides an opportunity to enhance the degassing of groundwaters, thereby facilitating carbonate precipitation prior to the arrival of drip water at the surface of the speleothem. This process, known as prior calcite precipitation (PCP), can also be favored by reductions in the partial pressure of $CO_2$ ($pCO_2$) within the cave due to ventilation. Ventilation typically occurs when external cooling provokes an exchange of air with the internal cave atmosphere, and when there is a reduction in the activity of vegetation leading to less respiration rates and $CO_2$ liberation. These processes reduced the $pCO_2$ in the soil and karstic system, supporting PCP.

- line 207: "lower respiration rates", I would clarify that soil or ecosystem respiration rates are mean here, as it comes a bit abruptly.

Respiration rates without any allusion to vegetation or soil indeed sound rare. We hope that this new sentence, together with the information provided in the preceding paragraph, will contribute to a more comprehensive understanding of the term.

Line 205: PCP can also be enhanced by higher dripwater saturation states even with constant drip rates (Stoll *et al.*, 2012), but the observation of maximum PCP during cold glacial periods characterized by lower soil $CO_2$ due to diminished microbial activity and vegetation respiration rates and likely lower initial dripwater saturation suggests that elevated glacial PCP is caused by slower drip rates and driven by lower infiltration and lower precipitation-evapotranspiration.

- line 209: maybe clarify that the higher initial d13C at low soil pCO2 is because of the proportionally higher concentration of atmospheric CO2 (with more positive d13C) in the soil gas?

We incorporate additional information in the text about this aspect, as you propose.

Line 209: Speleothem $\delta^{13}C$ can also reflect hydroclimate variations. The initial $\delta^{13}C$ of the drip water reflects a mixing signal derived from the carbon isotopic composition signatures of limestone, atmospheric $CO_2$, and the biogenic soil gas proceeding from both the microbial activity and the vegetation respiration (Hellstrom *et al.*, 1998). Soil autotropic and heterotrophic respiration typically have lower $\delta^{13}C$ values. Hence, cold and dry conditions can depress soil respiration rates and contribute to an increase in the $\delta^{13}C$. Likewise, the concentration of atmospheric $CO_2$ (with higher $\delta^{13}C$) would be proportionally higher than the relatively low biogenic carbon signature in the soil gas, elevating even more the $\delta^{13}C$ values. Is for that reason that many mid-latitude speleothem studies interpret high $\delta^{13}C$ as reflecting reduced soil/vegetation activity (Genty *et al.,* 2013; Moreno *et al.,* 2010; Bartolomé *et al.,* 2015; Pérez-Mejias *et al.*, 2017). Additionally, slowed drip rates during dry periods may contribute to greater PCP, which also causes dripwater to evolve to higher $\delta^{13}C$ due to the gradual loss of the light isotope. Thus, via these two processes (initial dripwater changes and PCP) dry conditions may contribute to higher speleothem $\delta^{13}C$.

Hellstrom J., McCulloch M. and Stone J. (1998) A detailed 31,000-year record of climate and vegetation change, from the isotope geochemistry of two New Zealand speleothems. Quat. Res. 50, 167–178.

- Discussion around line 280: I wonder how comparable the magnitude and structures of the events shown really are, since these are all qualitative proxy records (and not all the same proxy either), and not quantitatively reflecting processes? I would phrase more conservatively, since we don't know exactly how sensitive the different proxies react to local drivers and the climatic perturbations they record.

We made this change to adopt a more conservative approach in interpreting each record's sensitivity to climate fluctuations. We revised the sentence to:

Line 283: The Tenaghi Philippon age model relies on orbital tuning, assuming a constant lag of one thousand years in the orbital alignment, a circumstance that may account for certain discrepancies with the absolutely dated speleothem record. Aside from the age biases during deglaciations, both records exhibit proxy fluctuations associated with glacial/interglacial cycles as well as stadial/interstadial structures with a strong precession signal.

- line 286: It is not clear to me what the influence of precession on Mediterranean oceanography is. I think it would be good to clarify with another sentence.

In this section, we introduce some studies that demonstrate how precession is intimately related to the precipitation-evaporation balance of the Mediterranean basin. And how Mediterranean precipitation, to some extent in line with monsoons, can similarly influence ocean processes. To simplify this, we modified the introductory sentence and led the paragraph as follows:

Line 286: Precession has a major effect on the Mediterranean Sea due to its ability to disrupt the evaporation-precipitation balance of the sea basin. This influence has been documented through several proxy records that demonstrate modifications in oceanographic and sedimentary processes. For instance, marine sediment records close to the Strait of Gibraltar recorded changes in the Mediterranean Outflow Water strength (MOW), mainly driven by precession-controlled fluctuations in the Mediterranean hydrologic budget (Nichols *et al.,* 2020; Sierro *et al.*, 2020), and marine sediments from the Balearic Sea documented carbonate cyclicities due to the control of the W Mediterranean cyclogenetic mechanisms (Ochoa *et al.,* 2018). The precession control on the African monsoon is well known, which indirectly also affected the eastern Mediterranean water properties, through river runoff along the North African coast, changing drastically the $\delta^{18}O_{sw}$ composition and allowing sapropel formations (Rohling *et al.,* 2014; Grant *et al.*, 2016). Several studies highlight that the timing of enhanced summer monsoon corresponds with maximum Mediterranean autumn/winter storm track precipitation (Kutzbach *et al.*, 2014; Toucanne *et al.*, 2015; Wagner *et al.*, 2019). The Mg/Ca-RAT record also reveals this concordance, where western Mediterranean enhanced precipitation, during interglacial/interstadial periods of precession minima, is consistently in line with higher summer monsoon intensities according to the Chinese speleothem records from Sanbao cave (Fig. 3) (Cheng *et al.*, 2016).

- line 392: Again, two different terms for isotopic trends are used ("light" and "negative"). I would stick to one and use it for both records.

As commented before, we corrected and used the terms 'more negative/more positive' values for planktic foraminifera isotopes, their trends, and anomalies.

- line 417: This sentence in the conclusions highlighted to me that the discussion did not highlight enough how large the discrepancies between tuned and absolute chronologies are. This is a really important finding and I think it should be brought out more in the discussion.

Following your advice, we add more discussion about the discrepancies between the published and tuned chronologies in this study within section '3.2 Implications for Marine Chronologies'.

Line 399: Particularly, the adjusted TIV exhibits a temporal discrepancy of ~7.4 kyr compared to the established chronology, and the mean age uncertainty of the RAT speleothem throughout this termination is less than this difference, approximately

±3.4 kyr. In contrast, the TIII aligns well with previously published chronologies due to a minor time discrepancy, approximately ~3 kyr, similar to the RAT average age uncertainty of ±1.5 kyr. In the case of TIII.a, the chronological discrepancy is even more significant, reaching ~11 kyr, with average errors of the RAT estimated to be around ±1 kyr.

Figures:
- Figure 1: I find it difficult to understand from this figure and the caption which sites are the ones the data in this study comes from. Maybe add spell this detail out in the caption, and also consider different symbols or symbol sizes for the different "categories".

Stars are added to emphasize the records of this study.

[Figure]

- Figure 3: In the top bar with the ages, the colours are quite difficult to distinguish, particularly Ejulve and Corchia. I would also consider making the symbols just slightly larger and the error bars slightly thicker (I know there is not a lot of space as the plot is dense) to improve readability. I think it could also be nice, but I understand if this is not possible due to the denseness of the plot, if the d18O and d13C records of RAT could be included.

We decided not to include the RAT isotopic records in this plot because it is quite dense but also because our focus in this figure is to discuss the clear precision signal of Mg/Ca and contextualize it with precipitation patterns in the Mediterranean context (section 3.2). Allusions to the isotope records in the main text point to Figure 2 throughout the results section, while Figure 4 is dedicated to detailing the deglaciations and to highlight the significance of the oxygen record. Hence, we believe that it is not essential to add the isotopic records in Fig. 3. However, as you suggested, we made some changes to better distinguish speleothem absolute age errors.

[Figure]

- Figure 5: The line for the interglacial acme is difficult to see on the plot. In the caption, I would clarify "the d18O freshening structures in Iberia" (line 374).

We changed the figure for the next one and added 'in Iberia' in the figure caption.

---

## Author Comment (AC3)

**RC1**: Anonymous Referee #1, 21 Aug 2024

Torner et al use a speleothem record from the Western Mediterranean region to provide new absolute age constraints to the glacial terminations of TIV, TIII and TIIIa. They use speleothem proxy data to provide additional hydroclimatic context to these time periods. And provide extensive discussion on how this record can be used

- to evaluate orbital forcing on Terminations,
- to evaluate the differences between millennial structure of the different Terminations,
- to provide absolute age control to marine records through tie points
- and to provide a fuller picture of the sequence of climate events during Terminations.

It would be great to see this article published. I have only a few comments that would hopefully make it easier to garner more information from the article.

We sincerely appreciate your review of the article. We are pleased to know that you found it interesting enough to be considered for publication. We greatly value your observations and corrections, which will certainly improve the clarity of the manuscript. Below, we have addressed your comments.

Comments:
- In the introduction, please can you add just a sentence or so on why it is relevant to study Terminations in the context of current climate change

We will address this point after the first paragraph. We propose to add these sentences:

i.e Line 53: These feedbacks responsible for past deglacial warming can also play a role in the ongoing situation of climate change. The increase in atmospheric $CO_2$ due to anthropogenic emissions, the reduction in earth albedo resulting from ice melting, and even the weakening of the AMOC are currently in operation (IPCC, 2023; Boers, 2021). Studying past glacial terminations provides unique examples for elucidating the role that these internal components on Earth's climate system had during past periods of global climate warming, a test ground for better contextualizing current climate change.

Boers, N. Observation-based early-warning signals for a collapse of the Atlantic Meridional Overturning Circulation. *Nat. Clim. Chang.* **11**, 680–688 (2021). https://doi.org/10.1038/s41558-021-01097-4.

IPCC, 2023: *Climate Change 2023: Synthesis Report. Contribution of Working Groups I, II and III to the Sixth Assessment Report of the Intergovernmental Panel on Climate Change* [Core Writing Team, H. Lee and J. Romero (eds.). IPCC, Geneva, Switzerland, 184 pp., doi: 10.59327/IPCC/AR6-9789291691647, 2023.

- Lines 65-70: It would be great if any speleothem could truly have an unambiguous climatic interpretation! It would be better to phrase this as 'relatively unambiguous climate interpretation' or 'better understood climate interpretation'.

This is indeed a valuable consideration. Is true that there is no unambiguous interpretation of speleothem proxies. We have been too enthusiastic in this sentence. We appreciate your observation and will change the sentence for 'relatively unambiguous climate interpretation' as you suggested.

- Lines 105-110: Sub-heading 2.1 should be more appropriately named as maybe 'cave setting and regional hydroclimate' or so. All the cave description could come first followed by a description of regional hydroclimate. So the sentences at Line 122, 'The Murada cave....' Onward should move to the start of that section.

Thank you for your suggestion. We will implement this change and use 'Cave Settings and Regional Hydroclimate' as the sub-heading title. Additionally, as also recommended by Anonymous Referee 2, we will revise the structure of this section. Specifically, we will move the cave setting information to the beginning of the section and incorporate information from the other speleothem cave as well.

- Section sub-heading 2.2 should be Geochemical analyses.

We will correct that.

- Lines 140-145: Was the reversed age removed or included in the final age model generation by StalAge?

Of the 41 original samples, the age model was constructed using 39 samples. One of the excluded samples was a replicate used to evaluate consistency between different analysis dates, while the other was ignored because represented the most prominent reversal. Although the modeling process with StalAge considered all 39 samples, the final age model placed several dates at the limit of the model, with some exceeding the model's dating uncertainty. To clarify this, we will write 'with a few single dates exceeding the final modeled dating uncertainty' instead of 'with only a single reversal exceeding the dating uncertainty'. Furthermore, we will add visual support in Fig. 2 (attached at the end of the document)

Finally, this paragraph will be as follows, including also VALL 2 information:

i.e., Line 140-145: The speleothem depth-age models were performed with the R software and the StalAge package (Scholz, 2011), using 39 radiometric dates for RAT and 12 for VALL 2. The RAT speleothem provided a consistent chronology with a few single dates exceeding the final modeled dating uncertainty. It presents a remarkably precise age model for this period, with absolute uncertainties being even smaller in the youngest part of the speleothem (Fig. 2). The dates obtained

from the VALL 2 speleothem are approaching the limits of the U-series dating technique, thereby resulting in significant uncertainties, being even larger in the oldest section of the speleothem, where the growth rate is higher (Fig. C1). The growth rate ranges between 0.02-1.2 cm/kyr for VALL 2 and 0.07-1.7 cm/kyr for RAT.

- Lines 140-145: Add MIS and Termination numbers to this figure to make it easier to understand the text around Lines 190. Please can you also show the exact sample points as markers either in this figure or in the appendix. It would be good to see how many samples cover the slow growth periods.

Yes, we have changed this figure and incorporated your suggestions. The modified figure can be found at the end of this document.

- Lines 150-155: 'Concurrently measured' should instead be 'made on aliquots of the same powder sample'. Also remove the word estimated since these are simply results that have been measured and compared.

Yes, we will change both sentences as follows:

Stable isotopes and trace elements have been measured on aliquots of the same powder sample in both speleothems. The $\delta^{18}O$, $\delta^{13}C$, and Mg/Ca ratios have been compared in detail to evaluate their paleoclimatic signals.

- Lines 165-170: What is CCiT-UB?

The CCiT-UB is the acronym for the Scientific and Technological Centres of the University of Barcelona. This abbreviation was mentioned a few lines earlier (line 158), and for that reason, we are directly using the acronym here.

- Lines 265-270: Since so much of the discussion surrounds the sequence and duration of events, it would be great to have an idea of the uncertainties of these records, i.e. not just the speleothem record produced in this study, but also the other records being discussed in this figure. If the data are available, these could be plotted in the figure, if not, some estimates as a table or in the text would be great. Similarly, it would be good to see in the figure or have it noted in the text, if any event is represented by a single sample measurement. And the same for Figures 4 and 5.

You are absolutely correct in the necessity to clarify the uncertainties related to the data exposed. At the top of Figure 3, there are the dates with errors from Corchia and Ejulve records. Now, we also added dates with errors from the Sanbao cave record. The errors associated with the Tenagi Philippon record are not published, but they assume one kyr lag with the orbital parameters. To have a more comprehensive discussion we decided to add some additional information in the text too:

i.e. (line 267):  Other speleothem records from the Austrian Alps and Sardinia also reflect warming or more humid conditions along the MIS 7e responding to changes in the North Atlantic realm (Spötl *et al.*,2008; Columbu *et al.*, 2019; Wendt *et al.*, 2020). Considering that the typical dating error along the MIS 7e of the RAT record averages ~1.2 kyr, the discussed records present similar values, Sardinia (1.5 kyr) and Ejulve (average of 2 kyr). While the record from the Austrian Alps record reached exceptionally reduced errors of 0.3 kyr.

i.e. (line 282): The Tenaghi Philippon age model relies on orbital tuning, assuming a constant lag of one thousand years in the orbital alignment, a circumstance that may account for certain discrepancies with the absolutely dated speleothem record. Aside from the age biases during deglaciations, both records exhibit proxy fluctuations associated with glacial/interglacial cycles as well as stadial/interstadial structures with a strong precession signal.

Furthermore, in Figure 2 we add all the sample positions as symbols in the three proxy records to clarify that any of the further discussed events is represented by a single sample measurement. Additionally, a sentence will be added to provide further clarification:

i.e Line 156: Samples are spaced every 0.5 mm during glacial terminations to increase temporal resolution due to low growth rates, ensuring that events are captured by more than one measurement.

- There are some typos in the text so it could do with another check.

Thank you. We will do an additional check for typos and spelling.

**Figure 2**

[Figure]

**Figure 3**

[Figure]